# Rényi Differential Privacy of the Subsampled Shuffle Model in Distributed Learning

**Antonious M. Girgis**
UCLA
amgirgis@g.ucla.edu

**Deepesh Data**
UCLA
deepesh.data@gmail.com

**Suhas Diggavi**
UCLA
suhasdiggavi@ucla.edu

## Abstract

We study privacy in a distributed learning framework, where clients collaboratively build a learning model iteratively through interactions with a server from whom we need privacy. Motivated by stochastic optimization and the federated learning (FL) paradigm, we focus on the case where a small fraction of data samples are randomly sub-sampled in each round to participate in the learning process, which also enables privacy amplification. To obtain even stronger local privacy guarantees, we study this in the shuffle privacy model, where each client randomizes its response using a local differentially private (LDP) mechanism and the server only receives a random permutation (shuffle) of the clients' responses without their association to each client. The principal result of this paper is a privacy-optimization performance trade-off for discrete randomization mechanisms in this sub-sampled shuffle privacy model. This is enabled through a new theoretical technique to analyze the Rényi Differential Privacy (RDP) of the sub-sampled shuffle model. We numerically demonstrate that, for important regimes, with composition our bound yields significant improvement in privacy guarantee over the state-of-the-art approximate Differential Privacy (DP) guarantee (with strong composition) for sub-sampled shuffled models. We also demonstrate numerically significant improvement in privacy-learning performance operating point using real data sets. Despite these advances an open question is to bridge the gap between lower and upper privacy bounds in our RDP analysis.

## 1 Introduction

As learning moves towards the edge, there is a need to collaborate to build learning models[1], such as in federated learning [36, 44, 33]. In this framework, the collaboration is typically mediated by a server. In particular, we want to collaboratively build a learning model by solving an empirical risk minimization (ERM) problem (see (2) in Section 2). To obtain a model parametrized by $\theta$ using ERM, the commonly used mechanism is Stochastic Gradient Descent (SGD) [12]. However, one needs to solve this while enabling strong privacy guarantees on local data from the server, while also obtaining good learning performance, *i.e.,* a suitable privacy-learning performance operating point.

Differential privacy (DP) [18] is the gold standard notion of data privacy that gives a rigorous framework through quantifying the information leakage about individual training data points from the observed interactions. Though DP was originally proposed in a framework where data resides centrally [18], for distributed learning the more appropriate notion is of local differential privacy (LDP) [35, 17]. Here, each client randomizes its interactions with the server from whom the data is to be kept private (*e.g.,* see industrial implementations [23, 31, 16]). However, LDP mechanisms suffer

---

[1]This is because no client has access to enough data to build rich learning models locally and we do not want to directly share local data.

35th Conference on Neural Information Processing Systems (NeurIPS 2021).

from poor performance in comparison with the central DP mechanisms [17, 35, 32]. To overcome this, a new privacy framework using anonymization has been proposed in the so-called *shuffled model* [22, 25, 6, 26, 5, 15, 7, 8]. In the shuffled model, each client sends her private message to a secure shuffler that randomly permutes all the received messages before forwarding them to the server. This model enables significantly better privacy-utility performance by amplifying DP through this shuffling. Therefore, in this paper we consider the shuffle privacy framework for distributed learning.

In solving (2) using (distributed) gradient descent, each exchange leaks information about the local data, but we need as many steps as possible to obtain a good model; setting up the tension between privacy and performance. The goal is to obtain as many such interactions as possible for a given privacy budget. This is quantified through analyzing the privacy of the composition of privacy mechanisms. Abadi *et al.* [1] developed a framework for tighter analysis of such compositions, and this was later reformulated in terms of Rényi Differential Privacy (RDP) [37], and mapping this back to DP guarantee [38]. Therefore, studying RDP is important to obtaining strong composition privacy results, and is the focus of this paper.

In distributed (and federated) learning, a fraction of the data samples are sampled; for example, with random client participation and stochastic gradient descent (SGD), which can be written as

$$\theta_{t+1} \leftarrow \theta_t - \eta_t \frac{1}{|\mathcal{I}|} \sum_{i \in \mathcal{I}} \mathcal{R}(\nabla f_i(\theta_t)),$$

where $\mathcal{R}$ is the local randomization mechanism and $\mathcal{I}$ are the indices of the sampled data. This is a subsampled mechanism that enables another privacy amplification opportunity; which, in several cases, is shown to yield a privacy advantage proportional to the subsampling rate; see [35, 42]. The central technical question addressed in this paper is how to analyze the RDP of an arbitrary discrete mechanism for the subsampled shuffle privacy model. This enables us to answer the overall question posed in this paper, which is an achievable privacy-learning performance trade-off point for solving (2) in the shuffled privacy model for distributed learning (see Figure 1). Our contributions are:

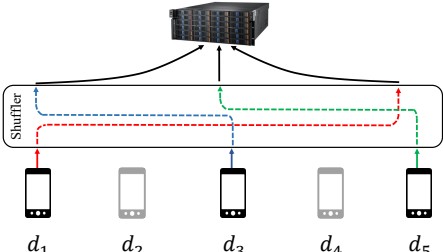

Figure 1: An iteration from the CLDP-SGD Algorithm, where 3 clients are randomly chosen at each iteration. Each client sends the private gradient $\mathcal{R}_p\left(g_t(d_i)\right)$ to the shuffler that randomly permutes the gradients before passing them to the server.

• We analyze the RDP of subsampled mechanisms in the shuffle framework by developing a novel bound applicable to any discrete $\epsilon_0$-LDP mechanism as a function of the RDP order $\lambda$, subsampling rate $\gamma$, the LDP parameter $\epsilon_0$, and the number of clients $n$; see Theorem 1. The bound is explicit and amenable to numerics, including all constants.[2] Furthermore, the bounds are valid for generic LDP mechanisms and *all* parameter regimes.[3] We also provide a lower bound for the RDP in Theorem 2. We prove our upper bound (Theorem 1) using the following novel analysis techniques: First, we reduce the problem of computing the RDP of sub-sampled shuffle mechanisms to the problem of computing ternary $|\chi|^{\alpha}$-DP [43] of shuffle (non sub-sampled) mechanisms; see Lemma 2. Then we reduce the computation of the ternary $|\chi|^{\alpha}$-DP of shuffle mechanisms for a *generic* triple of neighboring datasets to those that have a special structure (see Theorem 5) – this reduction step is one of the core technical results of this paper. Then we bound the ternary $|\chi|^{\alpha}$-DP of the shuffle mechanisms for triples of neighboring datasets having special structures by bounding the Pearson-Vajda divergence [43] using some concentration properties (see Theorem 6).

• Using the core technical result in Theorem 1, we analyze privacy-convergence trade-offs of the CLDP-SGD algorithm (see Algorithm 1) for Lipschitz convex functions in Theorem 3. This partially resolves an open question posed in [27], to extend their privacy analysis to RDP and significantly strengthening their privacy guatantees.

• Numerically, we save a factor $14\times$ in privacy ($\epsilon$) over the best known results for approximate DP for shuffling [24] combined with strong composition [34] for $T = 10^5, \gamma = 0.001, n = 10^6$, and a factor of $2.5\times$ better than the best known RDP for shuffling bound [29] combined with the sub-sampling result in [43]. Translating these to privacy-performance operating point in distributed

---

[2]As emphasized in [43], "in differential privacy, constants matter".

[3]Some of the best known approximate DP bounds for the shuffle model [7, 24] are restricted to certain parameter regimes in terms of $n, \delta, \epsilon_0$, etc.

optimization, over the MNIST data set with $\ell_\infty$ clipping we numerically show gains: For the same privacy budget of $\epsilon = 1.4$, we get a test performance of $80\%$ whereas using strong composition the test performance of [24] is $70\%$; furthermore, we achieves $90\%$ accuracy with the total privacy budget $\epsilon = 2.91$, whereas, [24] (with strong composition) achieves the same accuracy with a total privacy budget of $\epsilon = 4.82$. See Section 4 and the supplementary material for more results.

**Related work:** We give a more complete literature review in Appendix A, and focus here on the works that are closest to the results presented in this paper.

*Private optimization in the shuffled model:* Recently, [21] and [27, 28] have proposed differentially private SGD algorithms for federated learning, where at each iteration, each client applies an LDP mechanism on the gradients with the existence of a secure shuffler between the clients and the central server. However, the privacy analyses in these works developed approximate DP using advanced composition theorems for DP (*e.g.,* [20, 34]), which are known to be loose for composition [1]. To the best of our knowledge, analyzing the private optimization framework using RDP and subsampling in the shuffled model is new to this paper.

*Subsampled RDP:* The works [38, 43, 45] have studied the RDP of subsampled mechanisms *without shuffling*. They demonstrated that this provides a tighter bound on the total privacy loss than the bound that can be obtained using the standard strong composition theorems. The RDP analysis of subsampled mechanisms in the shuffled privacy framework has not been studied before,[4] and is new to this paper. The RDP of the shuffled model was very recently studied in [29], but without incorporating subsampling, which poses new technical challenges, as directly bounding the RDP of subsampled shuffle mechanisms is non-trivial. We overcome this by reducing our problem of computing RDP to bounding the ternary $|\chi|^\alpha$-DP, and bounding the latter is a core technical contribution of our paper.

**Paper organization:** We give preliminaries and problem formulation in Section 2, main results (upper and lower bounds, and privacy-convergence tradeoff) in Section 3, numerical results in Section 4, proof of the upper bound in Section 5, and proof of the ternary DP of the shuffle model in Section 6. Omitted details/proofs from this paper are given in the supplementary material.

## 2 Preliminaries and Problem Formulation

We use several privacy definitions throughout this paper. Among these, the local and central differential privacy definitions are standard and we defer them to Appendix B. The other privacy definitions (Rényi DP and ternary $|\chi|^\alpha$-DP) are relatively less standard and we define them below.

We say that two datasets $\mathcal{D} = \{d_1, \ldots, d_n\} \in \mathcal{X}^n$ and $\mathcal{D}' = \{d_1', \ldots, d_n'\} \in \mathcal{X}^n$ are neighboring (and denoted by $\mathcal{D} \sim \mathcal{D}'$) if they differ in one data point, i.e., there exists an $i \in [n]$ such that $d_i \neq d_i'$ and for every $j \in [n], j \neq i$, we have $d_j = d_j'$.

**Definition 1** (($\lambda, \epsilon$)-RDP (Rényi Differential Privacy) [37]). A randomized mechanism $\mathcal{M} : \mathcal{X}^n \rightarrow \mathcal{Y}$ is said to have $\epsilon$-Rényi differential privacy of order $\lambda \in (1, \infty)$ (in short, ($\lambda, \epsilon(\lambda)$)-RDP), if for any neighboring datasets $\mathcal{D}, \mathcal{D}' \in \mathcal{X}^n$, the Rényi divergence of order $\lambda$ between $\mathcal{M}(\mathcal{D})$ and $\mathcal{M}(\mathcal{D}')$ is upper-bounded by $\epsilon(\lambda)$, *i.e.,*

$$D_\lambda(\mathcal{M}(\mathcal{D})||\mathcal{M}(\mathcal{D}')) = \frac{1}{\lambda - 1} \log \left( \mathbb{E}_{\theta \sim \mathcal{M}(\mathcal{D}')} \left[ \left( \frac{\mathcal{M}(\mathcal{D})(\theta)}{\mathcal{M}(\mathcal{D}')(\theta)} \right)^\lambda \right] \right) \leq \epsilon(\lambda), \qquad (1)$$

where $\mathcal{M}(\mathcal{D})(\theta)$ denotes the probability that $\mathcal{M}$ on input $\mathcal{D}$ generates the output $\theta$.

**Definition 2** ($\zeta$-Ternary $|\chi|^\alpha$-differential privacy [43]). A randomized mechanism $\mathcal{M} : \mathcal{X}^n \rightarrow \mathcal{Y}$ is said to have $\zeta$-ternary-$|\chi|^\alpha$-DP, if for any triple of mutually adjacent datasets $\mathcal{D}, \mathcal{D}', \mathcal{D}'' \in \mathcal{X}^n$ (*i.e.,* they mutually differ in the same location), the ternary-$|\chi|^\alpha$ divergence of $\mathcal{M}(\mathcal{D}), \mathcal{M}(\mathcal{D}'), \mathcal{M}(\mathcal{D}')$ is upper-bounded by $(\zeta(\alpha))^\alpha$ for all $\alpha \geq 1$ (where $\zeta$ is a function from $\mathbb{R}^+$ to $\mathbb{R}^+$), *i.e.,*

$$D_{|\chi|^\alpha} \left( \mathcal{M}(\mathcal{D}), \mathcal{M}(\mathcal{D}')||\mathcal{M}(\mathcal{D}'') \right) := \mathbb{E}_{\mathcal{M}(\mathcal{D}'')} \left[ \left| \frac{\mathcal{M}(\mathcal{D}) - \mathcal{M}(\mathcal{D}')}{\mathcal{M}(\mathcal{D}'')} \right|^\alpha \right] \leq (\zeta(\alpha))^\alpha .$$

---

[4]One naive approach is to plug in the RDP analysis of shuffle model [29] into the results of [43]; however, our direct analysis of subsampled mechanisms yields better results in several interesting regimes; see Section 4.

The ternary $|\chi|^{\alpha}$-DP was proposed in [43] to characterize the RDP of the sub-sampled mechanism without shuffling. In this work, we analyze the ternary $|\chi|^{\alpha}$-DP of the shuffled mechanism to bound the RDP of the sub-sampled shuffle model.

We can use the following result for converting the RDP guarantees of a mechanism to its central DP guarantees. To the best of our knowledge, this result gives the best conversion.

**Lemma 1** (From RDP to DP [13, 4]). *Suppose for any $\lambda > 1$, a mechanism $\mathcal{M}$ is $(\lambda, \epsilon(\lambda))$-RDP. Then, the mechanism $\mathcal{M}$ is $(\epsilon, \delta)$-DP, where $\delta > 0$ is arbitrary and $\epsilon$ is given by*

$$\epsilon = \min_{\lambda} \left( \epsilon(\lambda) + \frac{\log(1/\delta) + (\lambda - 1)\log(1 - 1/\lambda) - \log(\lambda)}{\lambda - 1} \right).$$

**Problem formulation:** We consider a distributed private learning setup comprising a set of $n$ clients, where the $i$th client has a data point $d_i$ drawn from a universe $\mathcal{X}$ for $i \in [n]$; see also Figure 1. Let $\mathcal{D} = (d_1, \ldots, d_n)$ denote the entire training dataset. The clients are connected to an untrusted server in order to solve the following empirical risk minimization (ERM) problem

$$\min_{\theta \in \mathcal{C}} \left( F(\theta, \mathcal{D}) := \frac{1}{n} \sum_{i=1}^{n} f(\theta, d_i) \right), \tag{2}$$

where $\mathcal{C} \subset \mathbb{R}^d$ is a closed convex set, and $f : \mathcal{C} \times \mathcal{D} \to \mathbb{R}$ is the loss function. Our goal is to construct a global learning model $\theta$ via stochastic gradient descent (SGD) while preserving privacy of individual data points in the training dataset $\mathcal{D}$ by providing strong DP guarantees.

We revisit the CLDP-SGD algorithm presented in [27] and described in Algorithm 1 to solve the ERM (2). In each step of CLDP-SGD, we choose uniformly at random a set $\mathcal{U}_t$ of $k \leq n$ clients out of $n$ clients. Each client $i \in \mathcal{U}_t$ computes and clips the $\ell_p$-norm of the gradient $\nabla_{\theta_t} f(\theta_t, d_i)$ to apply the LDP mechanism $\mathcal{R}_p$, where $\mathcal{R}_p : \mathcal{B}_p^d \to \{0, 1\}^b$ is an $\epsilon_0$-LDP mechanism when inputs come from an $\ell_p$-norm ball. In [27], the authors proposed different $\epsilon_0$-LDP mechanisms for general $\ell_p$-norm balls. After that, the shuffler randomly permutes the received $k$ gradients $\{\mathcal{R}_p(\tilde{\mathbf{g}}_t(d_i))\}_{i \in \mathcal{U}_t}$ and sends them to the server. Finally,

---

**Algorithm 1** $\mathcal{A}_{\text{cldp}}$: CLDP-SGD

**Input:** Datasets $\mathcal{D} = (d_1, \ldots, d_n)$, LDP privacy parameter $\epsilon_0$, gradient norm bound $C$, and learning rate schedule $\{\eta_t\}$.

1: **Initialize:** $\theta_0 \in \mathcal{C}$
2: **for** $t \in [T]$ **do**
3:     **Client sampling:** A random set $\mathcal{U}_t$ of $k$ clients is chosen.
4:     **for** clients $i \in \mathcal{U}_t$ **do**
5:         *Compute gradient:* $\mathbf{g}_t(d_i) \leftarrow \nabla_{\theta_t} f(\theta_t, d_i)$
6:         *Clip gradient:* $\tilde{\mathbf{g}}_t(d_i) \leftarrow \mathbf{g}_t(d_i) / \max\left\{1, \frac{\|\mathbf{g}_t(d_i)\|_p}{C}\right\}$
7:         Client $i$ sends $\mathcal{R}_p(\tilde{\mathbf{g}}_t(d_i))$ to the shuffler.
8:     **end for**
9:     **Shuffling:** The shuffler sends random permutation of $\{\mathcal{R}_p(\tilde{\mathbf{g}}_t(d_i)) : i \in \mathcal{U}_t\}$ to the server.
10:    **Aggregate:** $\overline{\mathbf{g}}_t \leftarrow \frac{1}{k} \sum_{i \in \mathcal{U}_t} \mathcal{R}_p(\tilde{\mathbf{g}}_t(d_i))$
11:    **Descent Step:** $\theta_{t+1} \leftarrow \prod_{\mathcal{C}}(\theta_t - \eta_t \overline{\mathbf{g}}_t)$, where $\prod_{\mathcal{C}}$ is the projection operator onto the set $\mathcal{C}$.
12: **end for**

**Output:** The model $\theta_T$ and the privacy parameters $\epsilon, \delta$.

---

the server takes the average of the received gradients and updates the parameter vector. Our main contribution in this work is to present a stronger privacy analysis of the CLDP-SGD algorithm by characterizing the RDP of the sub-sampled shuffle model.

## 3 Main Results

In this section, we present our main results. First, we characterize the RDP of the subsampled shuffle mechanism by presenting an upper bound in Theorem 1 and a lower bound in Theorem 2. We then present the privacy-convergence trade-offs of the CLDP-SGD Algorithm in Theorem 3.

Consider an arbitrary $\epsilon_0$-LDP mechanism $\mathcal{R}$, whose range is a discrete set $[B] = \{1, \ldots, B\}$ for some $B \in \mathbb{N} := \{1, 2, 3, \ldots\}$. Here, $[B]$ could be the whole of $\mathbb{N}$. Let $\mathcal{M}(\mathcal{D})$ be a subsampled shuffle mechanism defined as follows: First subsample $k \leq n$ clients of the $n$ clients (without replacement), where $\gamma = \frac{k}{n}$ denotes the sampling parameter. Each client $i$ out of the $k$ selected

clients applies $\mathcal{R}$ on $d_i$ and sends $\mathcal{R}(d_i)$ to the shuffler,[5] who randomly permutes the received $k$ inputs and outputs the result. To formalize this, let $\mathcal{H}_k : \mathcal{Y}^k \to \mathcal{Y}^k$ denote the shuffling operation that takes $k$ inputs and outputs their uniformly random permutation. Let $\mathrm{samp}_k^n : \mathcal{X}^n \to \mathcal{X}^k$ denote the sampling operation for choosing a random subset of $k$ elements from a set of $n$ elements. We define the subsampled-shuffle mechanism as

$$\mathcal{M}(\mathcal{D}) := \mathcal{H}_k \circ \mathrm{samp}_k^n\left(\mathcal{R}(d_1), \ldots, \mathcal{R}(d_n)\right). \tag{3}$$

Observe that each iteration of Algorithm 1 can be represented as an output of the subsampled shuffle mechanism $\mathcal{M}$. Thus, to analyze the privacy of Algorithm 1, it is sufficient to analyze the privacy of a sequence of identical $T$ subsampled shuffle mechanisms, and then apply composition theorems.

**Histogram notation.** It will be useful to define the following notation. Since the output of $\mathcal{H}_k$ is a random permutation of the $k$ outputs of $\mathcal{R}$ (subsampling is not important here), the server cannot associate the $k$ messages to the clients; and the only information it can use from the messages is the histogram, i.e., the number of messages that give any particular output in $[B]$. We define a set $\mathcal{A}_B^k$ as

$$\mathcal{A}_B^k = \left\{ \boldsymbol{h} = (h_1, \ldots, h_B) : \sum_{j=1}^{B} h_j = k \right\}, \tag{4}$$

to denote the set of all possible histograms of the output of the shuffler with $k$ inputs. Therefore, we can assume, without loss of generality (w.l.o.g.), that the output of $\mathcal{M}$ is a distribution over $\mathcal{A}_B^k$.

Our main results for the RDP of the subsampled shuffled mechanism (defined in (3)) are given below. Our first result provides an upper bound (stated in Theorem 1 and proved in Section 5) and the second result provides a lower bound (stated in Theorem 2 and proved in Appendix D).

**Theorem 1** (Upper Bound). *For any $n \in \mathbb{N}$, $k \leq n$, $\epsilon_0 \geq 0$, and any integer $\lambda \geq 2$, the RDP of the subsampled shuffle mechanism $\mathcal{M}$ (defined in (3)) is upper-bounded by*

$$\epsilon(\lambda) \leq \frac{1}{\lambda - 1} \log \left( 1 + 4\binom{\lambda}{2} \gamma^2 \frac{(e^{\epsilon_0} - 1)^2}{\overline{k} e^{\epsilon_0}} + \sum_{j=3}^{\lambda} \binom{\lambda}{j} \gamma^j j \Gamma(j/2) \left( \frac{2(e^{2\epsilon_0} - 1)^2}{\overline{k} e^{2\epsilon_0}} \right)^{j/2} + \Upsilon \right),$$

*where $\overline{k} = \lfloor \frac{k-1}{2e^{\epsilon_0}} \rfloor + 1$, $\gamma = \frac{k}{n}$, and $\Gamma(z) = \int_0^\infty x^{z-1} e^{-x} dx$ is the Gamma function. The term $\Upsilon$ is given by $\Upsilon = \left( \left(1 + \gamma \frac{e^{2\epsilon_0} - 1}{e^{\epsilon_0}} \right)^\lambda - 1 - \lambda \gamma \frac{e^{2\epsilon_0} - 1}{e^{\epsilon_0}} \right) e^{-\frac{k-1}{8e^{\epsilon_0}}}.$*

**Theorem 2** (Lower Bound). *For any $n \in \mathbb{N}$, $k \leq n$, $\epsilon_0 \geq 0$, and any integer $\lambda \geq 2$, the RDP of the subsampled shuffle mechanism $\mathcal{M}$ (defined in (3)) is lower-bounded by*

$$\epsilon(\lambda) \geq \frac{1}{\lambda - 1} \log \left( 1 + \binom{\lambda}{2} \gamma^2 \frac{(e^{\epsilon_0} - 1)^2}{k e^{\epsilon_0}} + \sum_{j=3}^{\lambda} \binom{\lambda}{j} \gamma^j \left( \frac{(e^{2\epsilon_0} - 1)}{k e^{\epsilon_0}} \right)^j \mathbb{E}\left( m - \frac{k}{e^{\epsilon_0} + 1} \right)^j \right),$$

*where expectation is taken w.r.t. the binomial r.v. $m \sim \mathrm{Bin}(k, p)$ with parameter $p = \frac{1}{e^{\epsilon_0} + 1}$.*

Our CLDP-SGD algorithm and its privacy-convergence trade-offs (stated in Theorem 3 below) are given for a general local randomizer $\mathcal{R}_p$ (whose inputs comes from an $\ell_p$-ball for any $p \in [1, \infty]$) that satisfies the following conditions: (i) The randomized mechanism $\mathcal{R}_p$ is an $\epsilon_0$-LDP mechanism. (ii) The randomized mechanism $\mathcal{R}_p$ is unbiased, i.e., $\mathbb{E}[\mathcal{R}_p(\mathbf{x})|\mathbf{x}] = \mathbf{x}$ for all $\mathbf{x} \in \mathcal{B}_p(a)$, where $a$ is the radius of the ball $\mathcal{B}_p$. (iii) The output of the randomized mechanism $\mathcal{R}_p$ can be represented using $B \in \mathbb{N}^+$ bits. (iv) The randomized $\mathcal{R}_p$ has a bounded variance: $\sup_{\mathbf{x} \in \mathcal{B}_p(a)} \mathbb{E}\|\mathcal{R}_p(\mathbf{x}) - \mathbf{x}\|_2^2 \leq G_p^2(a)$, where $G_p^2$ is a function from $\mathbb{R}^+$ to $\mathbb{R}^+$.

Girgis et al. [27] proposed unbiased $\epsilon_0$-LDP mechanisms $\mathcal{R}_p$ for several values of norms $p \in [1, \infty]$ that require $b = \mathcal{O}(\log(d))$ bits of communication and satisfy the above conditions. In this paper, achieving communication efficiency is not our goal (though we also achieve that since the $\epsilon_0$-LDP mechanism $\mathcal{R}_p$ that we use takes values in a discrete set), as our main focus is on analyzing the RDP of the subsampled shuffle mechanism. If we use the $\epsilon_0$-LDP mechanism $\mathcal{R}_p$ from [27], we would also get similar gains in communication as were obtained in [27].

The privacy-convergence trade-off of our algorithm $\mathcal{A}_{\mathrm{cldp}}$ is given below.

---

[5]With a slight abuse of notation, in this paper we write $\mathcal{R}(d_i)$ to denote that $\mathcal{R}$ takes as its input the gradient computed on $d_i$ using the current parameter vector.

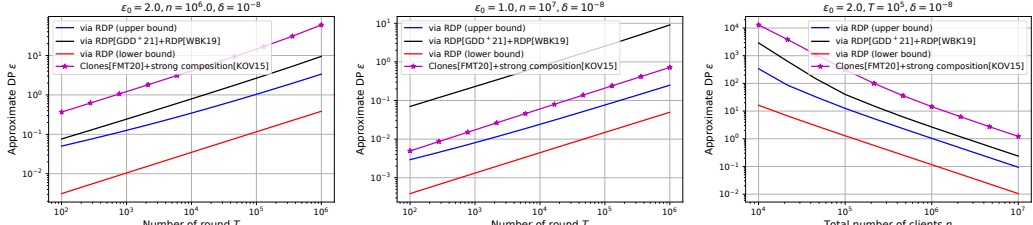

(a) Approx. DP as a function of $T$ for $\epsilon_0 = 2, \gamma = 0.001, n = 10^6$
(b) Approx. DP as a function of $T$ for $\epsilon_0 = 1, \gamma = 0.001, n = 10^7$
(c) Approx. DP as a function of $n$ for $\epsilon_0 = 2, \gamma n = 10^3, T = 10^5$

Figure 2: Comparison of several bounds on the Approximate $(\epsilon, \delta)$-DP for composition of a sequence of subsampled shuffle mechanisms for $\delta = 10^{-8}$: (i) Approximate DP obtained from our upper bound on the RDP in Theorem 1 (blue); (ii) Approximate DP obtained from our lower bound on the RDP in Theorem 2 (red); (iii) Approximate DP obtained from the upper bound on the RDP given in [29] with RDP amplification by subsampling from [43] (black); and (iv) Applying the strong composition theorem [34] after getting the Approximate DP of the shuffled model given in [24] with subsampling [42] (magenta).

**Theorem 3** (Privacy-Convergence tradeoffs). *Let the set $\mathcal{C}$ be convex with diameter $D$ and the function $f(\theta; .) : \mathcal{C} \times \mathcal{D} \to \mathbb{R}$ be convex and $L$-Lipschitz continuous with respect to the $\ell_g$-norm, which is the dual of the $\ell_p$-norm. Let $\theta^* = \arg\min_{\theta \in \mathcal{C}} F(\theta)$ denote the minimizer of the problem (2). For $\gamma = \frac{k}{n}$, if we run Algorithm $\mathcal{A}_{\text{cldp}}$ over $T$ iterations, then we have*

1. ***Privacy:*** *$\mathcal{A}_{\text{cldp}}$ is $(\epsilon, \delta)$-DP, where $\delta > 0$ is arbitrary and $\epsilon$ is given by*

$$\epsilon = \min_{\lambda} \left( T\epsilon(\lambda) + \frac{\log(1/\delta) + (\lambda - 1)\log(1 - 1/\lambda) - \log(\lambda)}{\lambda - 1} \right), \tag{5}$$

   *where $\epsilon(\lambda)$ is the RDP of the subsampled shuffle mechanism given in Theorem 1.*

2. ***Convergence:*** *If we run $\mathcal{A}_{\text{cldp}}$ with $\eta_t = \frac{D}{G\sqrt{t}}$, where $G^2 = \max\{d^{1-\frac{2}{p}}, 1\}L^2 + \frac{G_p^2(L)}{\gamma n}$, we get*

$$\mathbb{E}[F(\theta_T)] - F(\theta^*) \leq \mathcal{O}\left( \frac{DG\log(T)}{\sqrt{T}} \right).$$

The proof outline of Theorem 3 is as follows: Note that $\mathcal{A}_{\text{cldp}}$ is an iterative algorithm, where in each iteration we use the subsampled shuffle mechanism as defined in (3), for which we have computed the RDP guarantees in Theorem 1. Now, for the privacy analysis of $\mathcal{A}_{\text{cldp}}$, we use the adaptive composition theorem from [37, Proposition 1] and then use the RDP to DP conversion given in Lemma 1. For the convergence analysis, we use a standard non-private SGD convergence result and compute the required parameters for that. See Appendix F for a complete proof of Theorem 3.

**Remark 1.** Note that our convergence bound is affected by the variance of the $\epsilon_0$-LDP mechanism $\mathcal{R}_p$. For example, when $f$ is $L$-Lipschitz continuous w.r.t. the $\ell_2$-norm, we can use the LDP mechanism $\mathcal{R}_2$ proposed in [11] that has variance $G_2^2(L) = 14L^2 d \left( \frac{e^{\epsilon_0}+1}{e^{\epsilon_0}-1} \right)^2$; and when $f$ is $L$-Lipschitz continuous w.r.t. the $\ell_1$-norm or $\ell_\infty$-norm, we can use the LDP mechanisms $\mathcal{R}_\infty$ or $\mathcal{R}_1$, respectively, proposed in [27] that have variances $G_\infty^2(L) = L^2 d^2 \left( \frac{e^{\epsilon_0}+1}{e^{\epsilon_0}-1} \right)^2$ and $G_1^2(L) = L^2 d \left( \frac{e^{\epsilon_0}+1}{e^{\epsilon_0}-1} \right)^2$, respectively. By plugging these variances $G_p^2(L)$ (for $p = 1, 2, \infty$) into Theorem 3, we get the convergence rate of the $L$-Lipschitz continuous loss function w.r.t. the $\ell_p$-norm (for $p = \infty, 2, 1$).

**Remark 2.** The privacy parameter in (5) is not in a closed form expression and could be obtained by solving an optimization problem. However, we numerically compute it for several interesting regimes of parameters in our numerical experiments; see Section 4 for more details.

## 4 Numerical Results

In this section, we present numerical experiments to show the performance of our bounds on RDP of the subsampled shuffle mechanism and its usage for getting approximate DP of Algorithm 1 for training machine learning models.

**Composition of a sequence of subsampled shuffle models:** In Figure 2, we plot several bounds on the approximate $(\epsilon, \delta)$-DP for a composition of $T$ mechanisms $(\mathcal{M}_1, \ldots, \mathcal{M}_T)$, where $\mathcal{M}_t$ is

a subsampled shuffle mechanism for $t \in [T]$. In all our experiments reported in Figure 2, we fix $\delta = 10^{-8}$. We observe that our new bound on the RDP of the subsampled shuffle mechanism achieves a significant saving in total privacy $\epsilon$ compared to the state-of-the-art. For example, we save a factor of $14\times$ compared to the bound on DP [24] with strong composition theorem [34] and $2.5\times$ compared to the bound on the RDP given in [29] with subsampled RDP [43] in computing the overall privacy parameter $\epsilon$ for number of iterations $T = 10^5$, subsampling parameter $\gamma = 0.001$, LDP parameter $\epsilon_0 = 2$, and number of clients $n = 10^6$. We observe in Figure 2b that the bound given in [24] with the strong composition theorem [34] behaves better than the bound on the RDP [29] with subsampled RDP bound [43] when the number of subsampled clients per iteration is equal to $k = \gamma n = 10^4$; however, our bound beats both of them.[6] In Figure 2c, we fix the number of subsampled clients per iteration to be $k = \gamma n = 10^3$, and hence, the subsampling parameter $\gamma$ varies with $n$.

**Distributed private learning:** We numerically evaluate the proposed privacy-learning performance on training machine learning models. We consider the standard MNIST handwritten digit dataset that has $60,000$ training images and $10,000$ test images. We train a simple neural network that was also used in [21, 39] and described in Table 1. This model has $d = 13,170$ parameters and achieves an accuracy of $99\%$ for non-private, uncompressed vanilla SGD. We assume that we have $n = 60,000$ clients, where each client has one sample. At each step of the Algorithm 1, we choose uniformly at random $10,000$ clients, where each client clips the $\ell_\infty$-norm of the gradient with clipping parameter $C = 1/100$ and applies the $\mathcal{R}_\infty$ $\epsilon_0$-LDP mechanism proposed in [27] with $\epsilon_0 = 1.5$. We run Algorithm 1 with $\delta = 10^{-5}$ for 200 epochs, with learning rate $\eta = 0.3$ for the first 70 epochs, and then decrease it to $0.18$ in the remaining epochs.

| Layer | Parameters |
|---|---|
| Convolution | 16 filters of $8 \times 8$, Stride 2 |
| Max-Pooling | $2 \times 2$ |
| Convolution | 32 filters of $4 \times 4$, Stride 2 |
| Max-Pooling | $2 \times 2$ |
| Fully connected | 32 units |
| Softmax | 10 units |

Table 1: Model architecture for MNIST

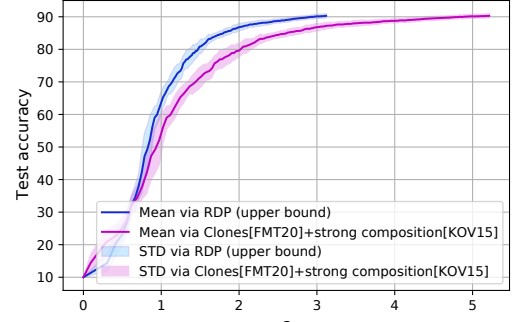

Figure 3: Privacy-Utility trade-offs on the MNIST dataset with $\ell_\infty$-norm clipping.

Figure 3 plots the mean and the standard deviation of privacy-accuracy trade-offs averaged over 10 runs. For our privacy analysis, the total privacy budget is computed by optimizing over RDP order $\lambda$ using our upper bound given in Theorem 1. For privacy analysis of [24], we first compute the privacy amplification by shuffling numerically given in [24]; then we compute its privacy obtained when amplified via subsampling [42]; and finally we use the strong composition theorem [34] to obtain the central privacy parameter $\epsilon$.

We observe that we achieve an accuracy of $80\%(\pm 1.8)$ with a total privacy budget of $\epsilon = 1.4$ using our new privacy analysis, whereas, [24] achieves an accuracy of only $70.7\%(\pm 2.1)$ with the same privacy budget of $\epsilon = 1.4$ using the standard composition theorems. Furthermore, we can see that we achieves accuracy $90\%(\pm 0.5)$ with total privacy budget $\epsilon = 2.91$ using our new privacy analysis, whereas, [24] (together with the standard strong composition theorem) achieves the same accuracy with a total privacy budget of $\epsilon = 4.82$.

# 5 Proof of Theorem 1: Upper Bound

For any dataset $\mathcal{D}_k = (d_1, \ldots, d_k) \in \mathcal{X}^k$ containing of $k$ data points, we define a shuffle mechanism $\mathcal{M}_{sh}(\mathcal{D}_k)$ as follows:

$$\mathcal{M}_{sh}(\mathcal{D}_k) = \mathcal{H}_k \left( \mathcal{R}\left(d_1\right), \ldots, \mathcal{R}\left(d_k\right) \right), \tag{6}$$

---

[6]In fact, there are several parameter regimes of great practical interest for which the results of [24] are not even valid; see Appendix G for more details on this, and also for more numerical comparisons.

where $\mathcal{H}_k$ takes $k$ inputs and outputs a uniformly random permutation of them. Recall from (3), for any dataset $\mathcal{D}_n = (d_1, \ldots, d_n) \in \mathcal{X}^n$ containing $n$ data points, the subsampled-shuffle mechanism is defined as $\mathcal{M}(\mathcal{D}) := \mathcal{H}_k \circ \mathrm{samp}_k^n(\mathcal{R}(d_1), \ldots, \mathcal{R}(d_n))$.

The proof of Theorem 1 consists of two steps. First, we bound the ternary-$|\chi|^\alpha$-DP of the shuffle mechanism $\mathcal{M}_{sh}$ (see Theorem 4), which is the main technical contribution in this proof. Then, using this, we bound the RDP of the subsampled shuffle mechanism $\mathcal{M}$.

**Theorem 4** ($\zeta$-ternary-$|\chi|^\alpha$-DP of the shuffle mechanism $\mathcal{M}_{sh}$). *For any integer $k \geq 2$, $\epsilon_0 > 0$, and all $\alpha \geq 2$, the $\zeta$-ternary-$|\chi|^\alpha$-DP of the shuffle mechanism $\mathcal{M}_{sh}$ is bounded by:*

$$\zeta(\alpha)^\alpha \leq \begin{cases} 4\frac{(e^{\epsilon_0}-1)^2}{\overline{k}e^{\epsilon_0}} + (e^{\epsilon_0} - e^{-\epsilon_0})^\alpha e^{-\frac{k-1}{8e^{\epsilon_0}}} & \text{if } \alpha = 2, \\ \alpha\Gamma(\alpha/2)\left(\frac{2(e^{2\epsilon_0}-1)^2}{\overline{k}e^{2\epsilon_0}}\right)^{\alpha/2} + (e^{\epsilon_0} - e^{-\epsilon_0})^\alpha e^{-\frac{k-1}{8e^{\epsilon_0}}} & \text{otherwise,} \end{cases} \tag{7}$$

*where $\overline{k} = \lfloor \frac{k-1}{2e^{\epsilon_0}} \rfloor + 1$ and $\Gamma(z) = \int_0^\infty x^{z-1}e^{-x}dx$ is the Gamma function.*

Theorem 4 is one of the core technical results of this paper, and we prove it in Section 6.

It was shown in [43, Proposition 16] that if a mechanism obeys $\zeta$-ternary-$|\chi|^\alpha$-DP, then its subsampled version (with subsampling parameter $\gamma$) will obey $\gamma\zeta$-ternary-$|\chi|^\alpha$-DP. Using that result, the authors then bounded the RDP of the subsampled mechanism in [43, Eq. (9)]. Adapting that result to our setting, we have the following lemma.

**Lemma 2** (From $\zeta$-ternary-$|\chi|^\alpha$-DP to subsampled RDP). *Suppose the shuffle mechanism $\mathcal{M}_{sh}$ obeys $\zeta$-ternary-$|\chi|^\alpha$-DP. For any $\lambda \geq 2, k \leq n$, RDP of the subsampled shuffle mechanism $\mathcal{M}$ (with subsampling parameter $\gamma = k/n$) is bounded by: $\epsilon(\lambda) \leq \frac{1}{\lambda-1}\log\left(1 + \sum_{\alpha=2}^\lambda \binom{\lambda}{\alpha}\gamma^\alpha\zeta(\alpha)^\alpha\right)$.*

Lemma 2 can be seen as a corollary to [43, Proposition 16 and Eq. (9)]. However, for completeness, we prove it in Appendix E.1. Substituting the bound on $\zeta(\alpha)$ from Theorem 4 into Lemma 2 together with some algebraic manipulation gives proves Theorem 1; see Appendix E.2 for details.

# 6 Proof of Theorem 4: Ternary $|\chi|^\alpha$-DP of the Shuffle Model

The proof has two main steps. In the first step, we reduce the problem of deriving ternary divergence for arbitrary neighboring datasets to the problem of deriving the ternary divergence for specific neighboring datasets, $\mathcal{D} \sim \mathcal{D}' \sim \mathcal{D}''$, where all elements in $\mathcal{D}$ are the same and $\mathcal{D}', \mathcal{D}''$ differ from $\mathcal{D}$ in one entry. In the second step, we derive the ternary divergence for the special neighboring datasets.

The specific neighboring datasets to which we reduce our general problem has the following form:

$$\mathcal{D}_{\text{same}}^m = \{(\mathcal{D}_m, \mathcal{D}_m', \mathcal{D}_m'') : \mathcal{D}_m = (d, \ldots, d, d) \in \mathcal{X}^m, \ \mathcal{D}_m' = (d, \ldots, d, d') \in \mathcal{X}^m, \text{ and}$$
$$\mathcal{D}_m'' = (d, \ldots, d, d'') \in \mathcal{X}^m, \text{ where } d, d', d'' \in \mathcal{X}\}, \tag{8}$$

Consider arbitrary neighboring datasets $\mathcal{D} = (d_1, \ldots, d_{k-1}, d_k)$, $\mathcal{D}' = (d_1, \ldots, d_{k-1}, d_k')$, and $\mathcal{D}'' = (d_1, \ldots, d_{k-1}, d_k'')$, each having $k$ elements. For any $m \in \{0, \ldots, k-1\}$, we define new neighboring datasets $\mathcal{D}_{m+1}^{(k)} = (d_k'', \ldots, d_k'', d_k)$, $\mathcal{D}_{m+1}'^{(k)} = (d_k'', \ldots, d_k'', d_k')$, and $\mathcal{D}_{m+1}''^{(k)} = (d_k'', \ldots, d_k'')$, each having $m+1$ elements. Observe that $(\mathcal{D}_{m+1}''^{(k)}, \mathcal{D}_{m+1}'^{(k)}, \mathcal{D}_{m+1}^{(k)}) \in \mathcal{D}_{\text{same}}^m$.

The first step of the proof is given in the following theorem.

**Theorem 5** (Reduction to the Special Case). *Let $q = \frac{1}{e^{\epsilon_0}}$. We have:*

$$\mathbb{E}_{\boldsymbol{h} \sim \mathcal{M}_{sh}(\mathcal{D}'')}\left[\left|\frac{\mathcal{M}_{sh}(\mathcal{D})(\boldsymbol{h}) - \mathcal{M}_{sh}(\mathcal{D}')(\boldsymbol{h})}{\mathcal{M}_{sh}(\mathcal{D}'')(\boldsymbol{h})}\right|^\alpha\right]$$
$$\leq \mathbb{E}_{m \sim \text{Bin}(k-1,q)}\left[\mathbb{E}_{\boldsymbol{h} \sim \mathcal{M}_{sh}(\mathcal{D}_{m+1}''^{(k)})}\left[\left|\frac{\mathcal{M}_{sh}(\mathcal{D}_{m+1}^{(k)})(\boldsymbol{h}) - \mathcal{M}_{sh}(\mathcal{D}_{m+1}'^{(k)})(\boldsymbol{h})}{\mathcal{M}_{sh}(\mathcal{D}_{m+1}''^{(k)})(\boldsymbol{h})}\right|^\alpha\right]\right]. \tag{9}$$

We know (by Chernoff bound) that the binomial r.v. is concentrated around its mean, which implies that the terms in the RHS of (9) that correspond to $m < (1 - \tau)q(k - 1)$ (we will take $\tau = 1/2$) will contribute in a negligible amount. Then we show that $E_m :=$

$\mathbb{E}_{\boldsymbol{h} \sim \mathcal{M}_{sh}(\mathcal{D}''^{(k)}_{m+1})}\left[\left|\frac{\mathcal{M}_{sh}(\mathcal{D}^{(k)}_{m+1})(\boldsymbol{h})-\mathcal{M}_{sh}(\mathcal{D}'^{(k)}_{m+1})(\boldsymbol{h})}{\mathcal{M}_{sh}(\mathcal{D}''^{(k)}_{m+1})(\boldsymbol{h})}\right|^{\alpha}\right]$ is a non-increasing function of $m$. These observation together imply that the RHS in (9) is approximately equal to $E_{(1-\tau)q(k-1)}$.

Since $E_m$ is precisely what is required to bound the ternary DP for the specific neighboring datasets, we have reduced the problem of computing the ternary DP for arbitrary neighboring datasets to the problem of computing ternary DP for specific neighboring datasets. The second step of the proof bounds $E_{(1-\tau)q(n-1)}$, which follows from the result below that holds for any $m \in \mathbb{N}$.

**Theorem 6** ($|\chi|^{\alpha}$-DP for special case). *For any $m \in \mathbb{N}$, integer $\alpha \geq 2$, and $(\mathcal{D}''_m, \mathcal{D}'_m, \mathcal{D}_m) \in \mathcal{D}^m_{same}$,*

$$\mathbb{E}_{\boldsymbol{h} \sim \mathcal{M}_{sh}(\mathcal{D}_m)}\left[\left|\frac{\mathcal{M}_{sh}(\mathcal{D}'_m)(\boldsymbol{h})-\mathcal{M}_{sh}(\mathcal{D}''_m)(\boldsymbol{h})}{\mathcal{M}_{sh}(\mathcal{D}_m)(\boldsymbol{h})}\right|^{\alpha}\right] \leq \begin{cases} 4\frac{(e^{\epsilon_0}-1)^2}{me^{\epsilon_0}} & \text{if } \alpha = 2, \\ \alpha\Gamma(\alpha/2)\left(\frac{2(e^{2\epsilon_0}-1)^2}{me^{2\epsilon_0}}\right)^{\alpha/2} & \text{otherwise.} \end{cases}$$

Missing details of how Theorem 4 follows from Theorems 5, 6 can be found in Appendix C.4.

*Proof sketch of Theorem 5.* Let $\boldsymbol{p}_i, i \in [k], \boldsymbol{p}'_k, \boldsymbol{p}''_k$ denote the distributions of $\mathcal{R}$ when the input data point is $d_i, d'_k, d''_k$, respectively. The main idea of the proof is the observation that each $\boldsymbol{p}_i$ can be written as a mixture distribution $\boldsymbol{p}_i = \frac{1}{e^{\epsilon_0}}\boldsymbol{p}''_k + \left(1 - \frac{1}{e^{\epsilon_0}}\right)\tilde{\boldsymbol{p}}_i$, where $\tilde{\boldsymbol{p}}_i$ is defined in terms of $\boldsymbol{p}_i, \boldsymbol{p}''_k$. So, instead of client $i \in [k-1]$ mapping its data point $d_i$ according to $\boldsymbol{p}_i$, we can view it as the client $i$ maps $d_i$ according to $\boldsymbol{p}''_k$ with probability (w.p.) $1/e^{\epsilon_0}$ and according to $\tilde{\boldsymbol{p}}_i$ w.p. $(1 - 1/e^{\epsilon_0})$. As a result, the number of clients that sample from the distribution $\boldsymbol{p}''_k$ follows a binomial distribution $\text{Bin}(k - 1, 1/e^{\epsilon_0})$. This allows us to write the distribution of $\mathcal{M}_{sh}$ when clients map their data points according to $\boldsymbol{p}_1, \ldots, \boldsymbol{p}_k, \boldsymbol{p}'_k, \boldsymbol{p}''_k$ as a convex combination of the distribution of $\mathcal{M}$ when clients map their data points according to $\tilde{\boldsymbol{p}}_1, \ldots, \tilde{\boldsymbol{p}}_{k-1}, \boldsymbol{p}_k, \boldsymbol{p}'_k, \boldsymbol{p}''_k$; see Lemma 4. Then using a joint convexity argument (see Lemma 3), we write the ternary divergence between the original triple of distributions of $\mathcal{M}_{sh}$ in terms of the same convex combination of the ternary divergence between the resulting triples of distributions of $\mathcal{M}_{sh}$ as in Lemma 4. Using a monotonicity argument (see Lemma 5), we can remove the effect of clients that do not sample from the distribution $\boldsymbol{p}''_k$ without decreasing the ternary divergence. By this chain of arguments, we have reduced the problem to the one involving the computation of ternary divergence only for the special form of neighboring datasets (as in Theorem 6), which proves Theorem 5. See Appendix C.1 for a complete proof. ∎

*Proof sketch of Theorem 6.* Consider $(\mathcal{D}''_m, \mathcal{D}'_m, \mathcal{D}_m) \in \mathcal{D}^m_{same}$ as in the statement of Theorem 6. First we observe that for any $\alpha \geq 1$ and any three distributions $p, q, r$ over the same domain, we can write $\mathbb{E}_r\left[\left|\frac{p-q}{r}\right|^{\alpha}\right] \leq 2^{\alpha-1}\left(\mathbb{E}_r\left[\left|\frac{p}{r}-1\right|^{\alpha}\right] + \mathbb{E}_r\left[\left|\frac{q}{r}-1\right|^{\alpha}\right]\right)$. This is a straight-forward application of the standard inequality $|x + y|^{\alpha} \leq 2^{\alpha-1}(|x|^{\alpha} + |y|^{\alpha})$ which holds for all $x, y \in \mathbb{R}$ and $\alpha \geq 1$. Now, by taking $p = \mathcal{M}_{sh}(\mathcal{D}'_m)$, $q = \mathcal{M}_{sh}(\mathcal{D}''_m)$, and $r = \mathcal{M}_{sh}(\mathcal{D}_m)$, we reduce the problem of computing the ternary $|\chi|^{\alpha}$-divergence (which we need to bound) to the problem of computing the Pearson-Vajda divergence [43], which we can write in terms of the $\alpha$-th absolute moment of the r.v. $X : \mathcal{A}^m_B \to \mathbb{R}$, defined as $X(\boldsymbol{h}) := \left(\frac{\mathcal{M}_{sh}(\mathcal{D}')(\boldsymbol{h})}{\mathcal{M}_{sh}(\mathcal{D}_m)(\boldsymbol{h})} - 1\right)$ for all $\boldsymbol{h} \in \mathcal{A}^m_B$ (where $\mathcal{D}' \in \{\mathcal{D}'_m, \mathcal{D}''_m\}$) and distributed according to $X(\boldsymbol{h}) \sim \mathcal{M}_{sh}(\mathcal{D}_m)(\boldsymbol{h})$. In [29], the authors have bounded the absolute moments of the r.v. $X(\boldsymbol{h})$ by showing that $X(\boldsymbol{h})$ is sub-Gaussian r.v. and using standard concentration results. See Appendix C.3 for a complete proof. ∎

## 7   Discussion

In this paper, we analyzed the Rényi differential privacy of the subsampled shuffle model by bounding the ternary $|\chi|^{\alpha}$-DP of the shuffle model. We numerically demonstrated the importance of our proposed bound, where we obtain a significant improvement over using the state-of-the-art in practical regimes. Furthermore, we used our privacy analysis to study the privacy-accuracy trade-offs on the MNIST dataset, where we obtained 90% accuracy with total privacy budget of $\epsilon = 2.91$, which is an improvement over an analysis yielding 4.82, using standard strong composition theorem.

Closing the gap (shown numerically) between our lower bound in Theorem 2 and the achievable upper bound in Theorem 1 is an important unresolved question. Another direction to explore would be to analyze the RDP of the subsampled shuffle model for different sub-sampling techniques such as Poisson subsampling [45], random check-in [9], or client self-sampling [30].

**Societal Impact.** Collaborative learning comes with significant societal risks of privacy violations, which is the main topic addressed in this paper. However, such learning is only as good as the data used for training, and if the data is not unbiased, this could lead to significant issues related to fairness and could also lead to societally undesirable outcomes. Such an issue is exacerbated when privacy is guaranteed on the data used for training, making a-priori fairness checks on data infeasible. This can be ameliorated by properly testing models finally obtained against fairness criteria and rejecting models that fail the test. This paper did not consider the issue of robustness to security, and this could also be an important societal issue in collaborative learning, where a small subset of users could insert malicious inputs to disrupt the learning process or worse bias the learned model covertly. This could also lead to negative outcomes. This issue of robustness to malicious participants has been studied in several papers, and incorporating this into the framework of the paper is an important future research topic.

## Acknowledgment

This work was supported in part by NSF grants #2007714 and #1955632 and a Google Faculty research award and an Amazon Research Award.

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
