# Supplementary Material

## A  Literature Review

We give the most relevant work related to the paper and review some of the main developments in differentially private learning below.

**Private Optimization:**  In [14], Chaudhuri et al. studied *centralized* privacy-preserving machine learning algorithms for convex optimization problem. In [10], Bassily et al. derived lower bounds on the empirical risk minimization under *central* differential privacy constraints. Furthermore, they proposed a differential privacy SGD algorithm that matches the lower bound for convex functions. In [1], the authors have generalized the private SGD algorithm proposed in [10] for non-convex optimization framework. In addition, the authors have proposed a new analysis technique, called moment accounting, to improve on the strong composition theorems to compute the central differential privacy guarantee for iterative algorithms. However, the works mentioned, [14, 10, 1], assume that there exists a trusted server that collects the clients' data. This motivates other works to design a distributed SGD algorithms, where each client perturbs her own data without needing a trusted server.

Distributed learning under local differential privacy (LDP) has studied in [2, 21, 27]. In [2] the authors proposed a communication-efficient algorithm for learning models under local differential privacy. In [21], the authors have proposed a distributed local-differential-privacy gradient descent algorithm, a newly proposed anonymization/shuffling framework [7] is used to amplify the privacy. In [27], the authors proposed communication efficient algorithms for general $\ell_p$-norm stetting under local differential privacy constraints, where they use recent results on amplification by shuffling to boost the privacy-utility trade-offs of the distributed learning algorithms.

**Shuffled privacy model:**  The shuffled model of privacy has been of significant recent interest [22, 25, 6, 26, 5, 15, 7, 8]. However, most of the existing works in literature [22, 7, 24] only characterize the approximate DP of the shuffled model. Recently, the authors in [29] proposed a novel bound on the RDP of the shuffled model, where they show that the RDP provides a significant saving in computing the total privacy budget for a composition of a sequence of shuffled mechanisms. However, the work [29] does not characterize the RDP of the subsampled shuffle mechanism. We can compute a bound on the RDP of the subsampled shuffle mechanism by combining the bound of the RDP of the shuffle mechanism in [29] with the bound of the subsampled RDP mechanism in [43]. However, we show numerically that our new bound on the subsampled shuffle mechanism outperforms this bound.

**Rényi differential privacy:**  The work of Abadi *et al.* [1] provided a new analysis technique to improve on the strong composition theorems. Inherently, this used Rényi divergence, and was later formalized in [37] which defined Rényi differential privacy (RDP). Several works [38, 43, 45] have shown that analyzing the RDP of subsampled mechanisms provides a tighter bound on the total privacy loss than the bound that can be obtained using the standard strong composition theorems. In this paper, we analyze the RDP of the subsampled shuffle model, where we can bound the approximate DP of a sequence of subsampled shuffle models using the transformation from RDP to approximate DP [1, 43, 13, 3]. We show that our RDP analysis provides a better bound on the total privacy loss of composition than the bound that can be obtained using the standard strong composition theorems and the bound that can be obtained by combining the RDP bound of the shuffle model in [29] with the subsampled RDP mechanism in [43].

## B  Additional Preliminaries

Here, we give local and central differential privacy definitions that we use throughout this work.

**Definition 3** (Local Differential Privacy - LDP [35])**.**  For $\epsilon_0 \geq 0$, a randomized mechanism $\mathcal{R} : \mathcal{X} \to \mathcal{Y}$ is said to be $\epsilon_0$-local differentially private (in short, $\epsilon_0$-LDP), if for every pair of inputs

$d, d' \in \mathcal{X}$, we have

$$\Pr[\mathcal{R}(d) \in \mathcal{S}] \le e^{\epsilon_0} \Pr[\mathcal{R}(d') \in \mathcal{S}], \qquad \forall \mathcal{S} \subset \mathcal{Y}. \tag{10}$$

Let $\mathcal{D} = \{d_1, \dots, d_n\}$ denote a dataset comprising $n$ points from $\mathcal{X}$. We say that two datasets $\mathcal{D} = \{d_1, \dots, d_n\}$ and $\mathcal{D}' = \{d'_1, \dots, d'_n\}$ are neighboring (and denoted by $\mathcal{D} \sim \mathcal{D}'$) if they differ in one data point, i.e., there exists an $i \in [n]$ such that $d_i \ne d'_i$ and for every $j \in [n], j \ne i$, we have $d_j = d'_j$.

**Definition 4** (Central Differential Privacy - DP [18, 19]). *For $\epsilon, \delta \ge 0$, a randomized mechanism $\mathcal{M} : \mathcal{X}^n \to \mathcal{Y}$ is said to be $(\epsilon, \delta)$-differentially private (in short, $(\epsilon, \delta)$-DP), if for all neighboring datasets $\mathcal{D}, \mathcal{D}' \in \mathcal{X}^n$ and every subset $\mathcal{S} \subseteq \mathcal{Y}$, we have*

$$\Pr[\mathcal{M}(\mathcal{D}) \in \mathcal{S}] \le e^{\epsilon_0} \Pr[\mathcal{M}(\mathcal{D}') \in \mathcal{S}] + \delta. \tag{11}$$

# C   Omitted Details from Section 6

## C.1   Proof of Theorem 5: Reduction to the Special Case

First, we prove the joint-convexity of the ternary $|\chi|^\alpha$-divergence as it is important in the following proof.

**Lemma 3** (Joint-convexity of the ternary $|\chi|^\alpha$-divergence). *For all $\alpha \ge 1$, the ternary-$|\chi|^\alpha$-divergence $\mathbb{E}\left[\left|\frac{P-Q}{R}\right|^\alpha\right]$ is jointly convex in $P, Q$ and $R$. In other words, if $P_a = aP_0 + (1-a)P_1$, $Q_a = aQ_0 + (1-a)Q_1$, and $R_a = aR_0 + (1-a)R_1$ for some $a \in [0,1]$, then the following holds*

$$\mathbb{E}_{R_a}\left[\left|\frac{P_a - Q_a}{R_a}\right|^\alpha\right] \le a\mathbb{E}_{R_0}\left[\left|\frac{P_0 - Q_0}{R_0}\right|^\alpha\right] + (1-a)\mathbb{E}_{R_1}\left[\left|\frac{P_1 - Q_1}{R_1}\right|^\alpha\right] \tag{12}$$

*Proof.* First, observe that $g(x, y) = |x - y|$ is jointly convex on $\mathbb{R}^2$, i.e., if $x_a = ax_0 + (1-a)x_1$ and $y_a = ay_0 + (1-a)y_1$, we have

$$
\begin{aligned}
|x_a - y_a| &= |a(x_0 - y_0) + (1-a)(x_1 - y_1)| \\
&\le a|x_0 - y_0| + (1-a)|x_1 - y_1|
\end{aligned} \tag{13}
$$

Let $f(x, y) = x^j/y^{j-1}$, which is jointly convex on $\mathbb{R}_+^2$ for $j \ge 1$; see [43, Lemma 20] for a proof. Thus, we get

$$\frac{|P_a - Q_a|^j}{R_a^{j-1}} \le \frac{(a|P_0 - Q_0| + (1-a)|P_1 - Q_1|)^j}{(aR_0 + (1-a)R_1)^{j-1}} \le a\frac{|P_0 - Q_0|^j}{R_0^{j-1}} + (1-a)\frac{|P_1 - Q_1|^j}{R_1^{j-1}}, \tag{14}$$

where the first inequality is obtained from (13) and the second inequality is obtained from the convexity of $f(x, y)$. ∎

Now, we prove Theorem 5. Our proof is an adaptation of the proof of [29, Theorem 4]. The difference comes from the fact that [29, Theorem 4] was for Renyi divergence, whereas, here we are working with ternary $|\chi|^\alpha$-divergence. This changes some details and we provide a full proof of Theorem 5 below.

Let $\boldsymbol{p}_i := (p_{i1}, \dots, p_{iB})$, $\boldsymbol{p}'_k := (p'_{k1}, \dots, p'_{kB})$, $\boldsymbol{p}''_k := (p''_{k1}, \dots, p''_{kB})$ denote the probability distributions over $\mathcal{Y}$ when the input to $\mathcal{R}$ is $d_i$, $d'_k$, and $d''_k$ respectively, where $p_{ij} = \Pr[\mathcal{R}(d_i) = j]$ for all $j \in [B]$ and $i \in [n]$. Let $\mathcal{P} = \{\boldsymbol{p}_i : i \in [k]\}$, $\mathcal{P}' = \{\boldsymbol{p}_i : i \in [k-1]\}\bigcup\{\boldsymbol{p}'_k\}$, and $\mathcal{P}'' = \{\boldsymbol{p}_i : i \in [k-1]\}\bigcup\{\boldsymbol{p}''_k\}$.

For $i \in [k-1]$, let $\mathcal{P}_{-i} = \mathcal{P} \setminus \{\boldsymbol{p}_i\}$ and also $\mathcal{P}_{-k} = \mathcal{P} \setminus \{\boldsymbol{p}_k\}$. Here, $\mathcal{P}, \mathcal{P}', \mathcal{P}''$ correspond to the datasets $\mathcal{D} = \{d_1, \dots, d_k\}, \mathcal{D}' = \{d_1, \dots, d_{k-1}, d'_k\}$, and $\mathcal{D}'' = \{d_1, \dots, d_{k-1}, d''_k\}$ respectively, and for any $i \in [k]$, $\mathcal{P}_{-i}$ corresponds to the dataset $\mathcal{D}_{-i} = \{d_1, \dots, d_{i-1}, d_{i+1}, \dots, d_k\}$.

For any collection $\mathcal{P} = \{\boldsymbol{p}_1, \dots, \boldsymbol{p}_k\}$ of $k$ distributions, we define $F(\mathcal{P})$ to be the distribution over $\mathcal{A}_B^k$ (which is the set of histograms on $B$ bins with $k$ elements as defined in (4)) that is induced

when every client $i$ (independent to the other clients) samples an element from $[B]$ accordingly to the probability distribution $\boldsymbol{p}_i$. Formally, for any $\boldsymbol{h} \in \mathcal{A}_B^k$, define

$$\mathcal{U}_{\boldsymbol{h}} := \left\{ (\mathcal{U}_1, \ldots, \mathcal{U}_B) : \mathcal{U}_1, \ldots, \mathcal{U}_B \subseteq [k] \text{ s.t. } \bigcup_{j=1}^{B} \mathcal{U}_j = [k] \text{ and } |\mathcal{U}_j| = h_j, \forall j \in [B] \right\}. \quad (15)$$

Note that for each $(\mathcal{U}_1, \ldots, \mathcal{U}_B) \in \mathcal{U}_{\boldsymbol{h}}, \mathcal{U}_j$ for $j = 1, \ldots, B$ denotes the identities of the clients that map to the $j$'th element in $[B]$ – here $\mathcal{U}_j$'s are disjoint for all $j \in [B]$. Note also that $|\mathcal{U}_{\boldsymbol{h}}| = \binom{k}{\boldsymbol{h}} = \frac{k!}{h_1! h_2! \ldots h_B!}$. It is easy to verify that for any $\boldsymbol{h} \in \mathcal{A}_B^k, F(\mathcal{P})(\boldsymbol{h})$ is equal to

$$F(\mathcal{P})(\boldsymbol{h}) = \sum_{(\mathcal{U}_1, \ldots, \mathcal{U}_B) \in \mathcal{U}_{\boldsymbol{h}}} \prod_{j=1}^{B} \prod_{i \in \mathcal{U}_j} p_{ij} \quad (16)$$

Similarly, we can define $F(\mathcal{P}'), F(\mathcal{P}''), F(\mathcal{P}_{-i}), F(\mathcal{P}'_{-i})$, and $F(\mathcal{P}''_{-i})$. Note that $F(\mathcal{P}), F(\mathcal{P}')$ and $F(\mathcal{P}'')$ are distributions over $\mathcal{A}_B^k$, whereas, $F(\mathcal{P}_{-i}), F(\mathcal{P}'_{-i})$, and $F(\mathcal{P}''_{-i})$ are distributions over $\mathcal{A}_B^{k-1}$. It is easy to see that $F(\mathcal{P}) = \mathcal{M}_{sh}(\mathcal{D}), F(\mathcal{P}') = \mathcal{M}_{sh}(\mathcal{D}')$, and $F(\mathcal{P}'') = \mathcal{M}_{sh}(\mathcal{D}'')$.

A crucial observation is that any distribution $\boldsymbol{p}_i$ can be written as the following mixture distribution:

$$\boldsymbol{p}_i = q \boldsymbol{p}_k'' + (1 - q) \tilde{\boldsymbol{p}}_i, \quad (17)$$

where $q = \frac{1}{e^{\epsilon_0}}$. The distribution $\tilde{\boldsymbol{p}}_i = [\tilde{p}_{i1}, \ldots, \tilde{p}_{iB}]$ is given by $\tilde{p}_{ij} = \frac{p_{ij} - q p_{kj}''}{1 - q}$, where it is easy to verify that $\tilde{p}_{ij} \geq 0$ and $\sum_{j=1}^{B} \tilde{p}_{ij} = 1$.

For any $\mathcal{C} \subseteq [k-1]$, define three sets $\mathcal{P}_{\mathcal{C}}, \mathcal{P}_{\mathcal{C}}'$, and $\mathcal{P}_{\mathcal{C}}''$ having $k$ distributions each, as follows:

$$\mathcal{P}_{\mathcal{C}} = \{\hat{\boldsymbol{p}}_1, \ldots, \hat{\boldsymbol{p}}_{k-1}\} \bigcup \{\boldsymbol{p}_k\}, \quad (18)$$

$$\mathcal{P}_{\mathcal{C}}' = \{\hat{\boldsymbol{p}}_1, \ldots, \hat{\boldsymbol{p}}_{k-1}\} \bigcup \{\boldsymbol{p}_k'\}, \quad (19)$$

$$\mathcal{P}_{\mathcal{C}}'' = \{\hat{\boldsymbol{p}}_1, \ldots, \hat{\boldsymbol{p}}_{k-1}\} \bigcup \{\boldsymbol{p}_k''\}, \quad (20)$$

where, for every $i \in [k-1], \hat{\boldsymbol{p}}_i$ is defined as follows:

$$\hat{\boldsymbol{p}}_i = \begin{cases} \boldsymbol{p}_k'' & \text{if } i \in \mathcal{C}, \\ \tilde{\boldsymbol{p}}_i & \text{if } i \in [k-1] \setminus \mathcal{C}. \end{cases} \quad (21)$$

In the following lemma, we show that $F(\mathcal{P}), F(\mathcal{P}')$, and $F(\mathcal{P}'')$ can be written as convex combinations of $\{F(\mathcal{P}_{\mathcal{C}}) : \mathcal{C} \subseteq [k-1]\}, \{F(\mathcal{P}_{\mathcal{C}}') : \mathcal{C} \subseteq [n-1]\}$, and $\{F(\mathcal{P}_{\mathcal{C}}'') : \mathcal{C} \subseteq [k-1]\}$, respectively, where for any $\mathcal{C} \subseteq [k-1], F(\mathcal{P}_{\mathcal{C}}), F(\mathcal{P}_{\mathcal{C}}')$. and $F(\mathcal{P}_{\mathcal{C}}'')$ can be computed analogously as in (16).

**Lemma 4** (Mixture Interpretation). *$F(\mathcal{P}), F(\mathcal{P}')$, and $F(\mathcal{P}'')$ can be written as the following convex combinations:*

$$F(\mathcal{P}) = \sum_{\mathcal{C} \subseteq [k-1]} q^{|\mathcal{C}|} (1-q)^{k-|\mathcal{C}|-1} F(\mathcal{P}_{\mathcal{C}}), \quad (22)$$

$$F(\mathcal{P}') = \sum_{\mathcal{C} \subseteq [k-1]} q^{|\mathcal{C}|} (1-q)^{k-|\mathcal{C}|-1} F(\mathcal{P}_{\mathcal{C}}'), \quad (23)$$

$$F(\mathcal{P}'') = \sum_{\mathcal{C} \subseteq [k-1]} q^{|\mathcal{C}|} (1-q)^{k-|\mathcal{C}|-1} F(\mathcal{P}_{\mathcal{C}}''), \quad (24)$$

*where $\mathcal{P}_{\mathcal{C}}, \mathcal{P}_{\mathcal{C}}', \mathcal{P}_{\mathcal{C}}''$ are defined in (18)-(21).*

We present a proof of Lemma 4 in Appendix C.2.

From Lemma 3 and Lemma 4, we get

$$\mathbb{E}_{\boldsymbol{h} \sim F(\mathcal{P}'')} \left[ \left| \frac{F(\mathcal{P})(\boldsymbol{h}) - F(\mathcal{P}')(\boldsymbol{h})}{F(\mathcal{P}'')(\boldsymbol{h})} \right|^{\alpha} \right]$$

$$\leq \sum_{\mathcal{C} \subseteq [k-1]} q^{|\mathcal{C}|} (1-q)^{k-|\mathcal{C}|-1} \, \mathbb{E}_{\boldsymbol{h} \sim F\left(\mathcal{P}_{\mathcal{C}}''\right)} \left[ \left| \frac{F(\mathcal{P}_{\mathcal{C}})(\boldsymbol{h}) - F(\mathcal{P}_{\mathcal{C}}')(\boldsymbol{h})}{F(\mathcal{P}_{\mathcal{C}}'')(\boldsymbol{h})} \right|^{\alpha} \right]. \quad (25)$$

For any $\mathcal{C} \subseteq [k-1]$, let $\widetilde{\mathcal{P}}_{[k-1]\setminus\mathcal{C}} = \{\tilde{\boldsymbol{p}}_i : i \in [k-1] \setminus \mathcal{C}\}$. With this notation, note that $\mathcal{P}_{\mathcal{C}} \setminus \widetilde{\mathcal{P}}_{[k-1]\setminus\mathcal{C}} = \{\boldsymbol{p}_k'', \ldots, \boldsymbol{p}_k''\} \bigcup \{\boldsymbol{p}_k\}, \mathcal{P}_{\mathcal{C}}' \setminus \widetilde{\mathcal{P}}_{[k-1]\setminus\mathcal{C}} = \{\boldsymbol{p}_k'', \ldots, \boldsymbol{p}_k''\} \bigcup \{\boldsymbol{p}_k'\}$, and $\mathcal{P}_{\mathcal{C}}'' \setminus \widetilde{\mathcal{P}}_{[k-1]\setminus\mathcal{C}} = \{\boldsymbol{p}_k'', \ldots, \boldsymbol{p}_k''\} \bigcup \{\boldsymbol{p}_k''\}$ are a triple of specific neighboring distributions, each containing $|\mathcal{C}| + 1$ distributions. In other words, if we define $\mathcal{D}_{|\mathcal{C}|+1}^{(k)} = (d_k'', \ldots, d_k'', d_k), \mathcal{D}_{|\mathcal{C}|+1}'^{(k)} = (d_k'', \ldots, d_k'', d_k')$, and $\mathcal{D}_{|\mathcal{C}|+1}''^{(k)} = (d_k'', \ldots, d_k'', d_k'')$, each having $(|\mathcal{C}| + 1)$ data points (note that $(\mathcal{D}_{|\mathcal{C}|+1}''^{(k)}, \mathcal{D}_{|\mathcal{C}|+1}'^{(k)}, \mathcal{D}_{|\mathcal{C}|+1}^{(k)}) \in \mathcal{D}_{\text{same}}^{|\mathcal{C}|+1}$), then the mechanisms $\mathcal{M}_{sh}(\mathcal{D}_{|\mathcal{C}|+1}^{(k)}), \mathcal{M}_{sh}(\mathcal{D}_{|\mathcal{C}|+1}'^{(k)})$, and $\mathcal{M}_{sh}(\mathcal{D}_{|\mathcal{C}|+1}''^{(k)})$ will have distributions $F(\mathcal{P}_{\mathcal{C}} \setminus \widetilde{\mathcal{P}}_{[k-1]\setminus\mathcal{C}}), F(\mathcal{P}_{\mathcal{C}}' \setminus \widetilde{\mathcal{P}}_{[k-1]\setminus\mathcal{C}})$, and $F(\mathcal{P}_{\mathcal{C}}'' \setminus \widetilde{\mathcal{P}}_{[k-1]\setminus\mathcal{C}})$, respectively.

Now, since $(\mathcal{D}_{|\mathcal{C}|+1}''^{(k)}, \mathcal{D}_{|\mathcal{C}|+1}'^{(k)}, \mathcal{D}_{|\mathcal{C}|+1}^{(k)}) \in \mathcal{D}_{\text{same}}^{|\mathcal{C}|+1}$, if we remove the effect of distributions in $\widetilde{\mathcal{P}}_{[k-1]\setminus\mathcal{C}}$ in the RHS of (25), we would be able to bound the RHS of (25) using the ternary $|\chi|^{\alpha}$-divergence for the special neighboring datasets in $\mathcal{D}_{\text{same}}^{|\mathcal{C}|+1}$. This is precisely what we will do in the following lemma and the subsequent corollary, where we will eliminate the distributions in $\widetilde{\mathcal{P}}_{[k-1]\setminus\mathcal{C}}$ in the RHS (25).

The following lemma holds for arbitrary triples $(\mathcal{P}, \mathcal{P}', \mathcal{P}'')$ of neighboring distributions $\mathcal{P} = \{\boldsymbol{p}_1, \ldots, \boldsymbol{p}_{k-1}, \boldsymbol{p}_k\}, \mathcal{P}' = \{\boldsymbol{p}_1, \ldots, \boldsymbol{p}_{k-1}, \boldsymbol{p}_k'\}$, and $\mathcal{P}'' = \{\boldsymbol{p}_1, \ldots, \boldsymbol{p}_{k-1}, \boldsymbol{p}_k''\}$, where we show that the ternary $|\chi|^{\alpha}$-divergence $\mathbb{E}_{\boldsymbol{h} \sim F(\mathcal{P}'')} \left[ \left| \frac{F(\mathcal{P})(\boldsymbol{h}) - F(\mathcal{P}')(\boldsymbol{h})}{F(\mathcal{P}'')(\boldsymbol{h})} \right|^{\alpha} \right]$ does not decrease when we eliminate a distribution $\boldsymbol{p}_i$ (i.e., remove the data point $d_i$ from the datasets) for any $i \in [k-1]$.

**Lemma 5** (Monotonicity). *For any $i \in [k-1]$, we have*

$$\mathbb{E}_{\boldsymbol{h} \sim F(\mathcal{P}'')} \left[ \left| \frac{F(\mathcal{P})(\boldsymbol{h}) - F(\mathcal{P}')(\boldsymbol{h})}{F(\mathcal{P}'')(\boldsymbol{h})} \right|^{\alpha} \right] \leq \mathbb{E}_{\boldsymbol{h} \sim F(\mathcal{P}_{-i}'')} \left[ \left| \frac{F(\mathcal{P}_{-i})(\boldsymbol{h}) - F(\mathcal{P}_{-i}')(\boldsymbol{h})}{F(\mathcal{P}_{-i}'')(\boldsymbol{h})} \right|^{\alpha} \right]. \quad (26)$$

*Proof.* This can be proved along the lines of the proof of [29, Lemma 5], which shows that $\mathbb{E}_{\boldsymbol{h} \sim F(\mathcal{P}')} \left[ \left( \frac{F(\mathcal{P})(\boldsymbol{h})}{F(\mathcal{P}')(\boldsymbol{h})} \right)^{\lambda} \right] \leq \mathbb{E}_{\boldsymbol{h} \sim F(\mathcal{P}_{-i}')} \left[ \left( \frac{F(\mathcal{P}_{-i})(\boldsymbol{h})}{F(\mathcal{P}_{-i}')(\boldsymbol{h})} \right)^{\lambda} \right]$ holds for all $i \in [k-1]$. This is a result about Renyi divergence, and the only property of the Renyi divergence that is used in the proof of [29, Lemma 5] is that $\mathbb{E}_{\boldsymbol{h} \sim F(\mathcal{P}')} \left[ \left( \frac{F(\mathcal{P})(\boldsymbol{h})}{F(\mathcal{P}')(\boldsymbol{h})} \right)^{\lambda} \right]$ is convex in $\boldsymbol{p}_i$ for any $i \in [k-1]$.

Note that Lemma 5 is about the ternary $|\chi|^{\alpha}$-divergence, and the required convexity about this follows from Lemma 3. So, following the proof of [29, Lemma 5] and using Lemma 3, proves Lemma 5. ∎

Now, for any given $\mathcal{C} \subseteq [k-1]$, by eliminating the distributions $\tilde{\boldsymbol{p}}_i$ in $\widetilde{\mathcal{P}}_{[k-1]\setminus\mathcal{C}} = \{\tilde{\boldsymbol{p}}_i : i \in [k-1] \setminus \mathcal{C}\}$ from $\mathcal{P}_C, \mathcal{P}_C'$, and $\mathcal{P}_C''$ (by repeatedly applying Lemma 5), we get that

$$\mathbb{E}_{\boldsymbol{h} \sim F\left(\mathcal{P}_{\mathcal{C}}''\right)} \left[ \left| \frac{F(\mathcal{P}_{\mathcal{C}})(\boldsymbol{h}) - F(\mathcal{P}_{\mathcal{C}}')(\boldsymbol{h})}{F(\mathcal{P}_{\mathcal{C}}'')(\boldsymbol{h})} \right|^{\alpha} \right]$$
$$\leq \mathbb{E}_{\boldsymbol{h} \sim \mathcal{M}_{sh}(\mathcal{D}_{m+1}''^{(k)})} \left[ \left| \frac{\mathcal{M}_{sh}(\mathcal{D}_{m+1}^{(k)})(\boldsymbol{h}) - \mathcal{M}_{sh}(\mathcal{D}_{m+1}'^{(k)})(\boldsymbol{h})}{\mathcal{M}_{sh}(\mathcal{D}_{m+1}''^{(k)})(\boldsymbol{h})} \right|^{\alpha} \right], \quad (27)$$

where $m = |\mathcal{C}|$. By substituting from (27) into (25) completes the proof of Theorem 5.

### C.2 Proof of Lemma 4

This can be proved along the lines of the proof of [29, Lemma 3], and we prove it below for completeness.

We only show (22); (23) and (24) can be shown similarly.

For convenience, for any $\mathcal{C} \subseteq [k-1]$, define

$$\mathcal{P}''_{|\mathcal{C}|,k} = \{\boldsymbol{p}''_k, \ldots, \boldsymbol{p}''_k\} \text{ with } |\mathcal{P}''_{|\mathcal{C}|,k}| = |\mathcal{C}|,$$
$$\widetilde{\mathcal{P}}_{[k-1]\backslash \mathcal{C}} = \{\tilde{\boldsymbol{p}}_i : i \in [k-1] \backslash \mathcal{C}\}.$$

With these notations, we can write $\mathcal{P}_{\mathcal{C}} = \mathcal{P}''_{|\mathcal{C}|,k} \bigcup \widetilde{\mathcal{P}}_{[k-1]\backslash \mathcal{C}} \bigcup \{\boldsymbol{p}_k\}$, and $\mathcal{P}'_{\mathcal{C}} = \mathcal{P}''_{|\mathcal{C}|,k} \bigcup \widetilde{\mathcal{P}}_{[k-1]\backslash \mathcal{C}} \bigcup \{\boldsymbol{p}'_k\}$, and $\mathcal{P}''_{\mathcal{C}} = \mathcal{P}''_{|\mathcal{C}|,k} \bigcup \widetilde{\mathcal{P}}_{[k-1]\backslash \mathcal{C}} \bigcup \{\boldsymbol{p}''_k\}$.

Note that $\boldsymbol{p}_i = q\boldsymbol{p}'_k + (1-q)\tilde{\boldsymbol{p}}_i$ for all $i \in [k-1]$. For any $i \in [k-1]$, define the following random variable $\widehat{\boldsymbol{p}}_i$:

$$\widehat{\boldsymbol{p}}_i = \begin{cases} \boldsymbol{p}''_k & \text{w.p. } q, \\ \tilde{\boldsymbol{p}}_i & \text{w.p. } 1-q. \end{cases}$$

Note that $\mathbb{E}[\widehat{\boldsymbol{p}}_i] = \boldsymbol{p}_i$. For any subset $\mathcal{C} \subseteq [k-1]$, define an event $\mathcal{E}_{\mathcal{C}} := \{\widehat{\boldsymbol{p}}_i = \boldsymbol{p}''_k \text{ for } i \in \mathcal{C} \text{ and } \widehat{\boldsymbol{p}}_i = \tilde{\boldsymbol{p}}_i \text{ for } i \in [k-1] \backslash \mathcal{C}\}$. Since $\widehat{\boldsymbol{p}}_1, \ldots, \widehat{\boldsymbol{p}}_{n-1}$ are independent random variables, we have $\Pr[\mathcal{E}_{\mathcal{C}}] = q^{|\mathcal{C}|}(1-q)^{n-|\mathcal{C}|-1}$.

Consider an arbitrary $\boldsymbol{h} \in \mathcal{A}_B^n$. Define a random variable $U(\mathcal{P})$ over $\mathcal{A}_B^k$ whose distribution is equal to $F(\mathcal{P})$.

$$\begin{aligned}
F(\mathcal{P})(\boldsymbol{h}) &= \Pr[U(\mathcal{P}) = \boldsymbol{h}] \\
&= \Pr[U(\boldsymbol{p}_1, \ldots, \boldsymbol{p}_{k-1}, \boldsymbol{p}_n) = \boldsymbol{h}] \\
&= \Pr\left[U\left(\mathbb{E}[\widehat{\boldsymbol{p}}_1], \ldots, \mathbb{E}[\widehat{\boldsymbol{p}}_{k-1}], \boldsymbol{p}_k\right) = \boldsymbol{h}\right] \\
&= \sum_{\mathcal{C} \subseteq [k-1]} \Pr[\mathcal{E}_{\mathcal{C}}] \Pr\left[U\left(\mathbb{E}[\widehat{\boldsymbol{p}}_1], \ldots, \mathbb{E}[\widehat{\boldsymbol{p}}_{k-1}], \boldsymbol{p}_k\right) = \boldsymbol{h} \mid \mathcal{E}_{\mathcal{C}}\right] \\
&\stackrel{(e)}{=} \sum_{\mathcal{C} \subseteq [k-1]} \Pr[\mathcal{E}_{\mathcal{C}}] \Pr\left[U\left(\mathcal{P}'_{|\mathcal{C}|,k} \bigcup \widetilde{\mathcal{P}}_{[k-1]\backslash \mathcal{C}} \bigcup \{\boldsymbol{p}_k\}\right) = \boldsymbol{h}\right] \\
&= \sum_{\mathcal{C} \subseteq [k-1]} \Pr[\mathcal{E}_{\mathcal{C}}] \Pr\left[U(\mathcal{P}_{\mathcal{C}}) = \boldsymbol{h}\right] \\
&= \sum_{\mathcal{C} \subseteq [k-1]} q^{|\mathcal{C}|}(1-q)^{k-|\mathcal{C}|-1} \Pr\left[U(\mathcal{P}_{\mathcal{C}}) = \boldsymbol{h}\right], \\
&= \sum_{\mathcal{C} \subseteq [k-1]} q^{|\mathcal{C}|}(1-q)^{k-|\mathcal{C}|-1} F(\mathcal{P}_{\mathcal{C}})(\boldsymbol{h}) \qquad (28)
\end{aligned}$$

where, $\mathcal{P}'_{|\mathcal{C}|,k}$ and $\widetilde{\mathcal{P}}_{[k-1]\backslash \mathcal{C}}$ in the RHS of (e) are defined in the statement of the claim.

Since the above calculation holds for every $\boldsymbol{h} \in \mathcal{A}_B^k$, we have proved (22).

### C.3    Proof of Theorem 6: Ternary $|\chi|^\alpha$-DP of the Special Case

First, we present the following standard inequality which is important to our proof.

**Lemma 6.** *Let $x, y \in \mathbb{R}$ be any two real numbers. Then, for all $j \geq 1$, we have*

$$|x+y|^j \leq 2^{j-1}\left(|x|^j + |y|^j\right). \qquad (29)$$

*Proof.* The proof is simple from the convexity of the function $f(x) = x^j$ for $j \geq 1$.

$$|x+y|^j = 2^j \left|\frac{x+y}{2}\right|^j \leq 2^j \left(\frac{|x|+|y|}{2}\right)^j \leq 2^j \left(\frac{|x|^j + |y|^j}{2}\right) = 2^{j-1}\left(|x|^j + |y|^j\right),$$

where the second inequality is obtained from the Jensen's inequality and the fact that the function $f(x) = x^j$ is convex on $\mathbb{R}^+$ for all $j \geq 1$. $\blacksquare$

From Lemma 6, we get the following corollary.

**Corollary 1.** Fix an arbitrary $m \in \mathbb{N}$ and consider any three mutually neighboring datasets $\mathcal{D}_m$, $\mathcal{D}'_m$, $\mathcal{D}''_m$, where $\mathcal{D}_m = (d, \ldots, d) \in \mathcal{X}^m$, $\mathcal{D}'_m = (d, \ldots, d, d') \in \mathcal{X}^m$ and $\mathcal{D}''_m = (d, \ldots, d, d'') \in \mathcal{X}^m$. The ternary $|\chi|^\alpha$-DP is bounded by

$$
\mathbb{E}_{\boldsymbol{h} \sim \mathcal{M}_{sh}(\mathcal{D}_m)} \left[ \left| \frac{\mathcal{M}_{sh}(\mathcal{D}'_m)(\boldsymbol{h}) - \mathcal{M}_{sh}(\mathcal{D}''_m)(\boldsymbol{h})}{\mathcal{M}_{sh}(\mathcal{D}_m)(\boldsymbol{h})} \right|^\alpha \right]
$$
$$
\leq 2^{\alpha-1} \left( \mathbb{E}_{\boldsymbol{h} \sim \mathcal{M}_{sh}(\mathcal{D}_m)} \left[ \left| \frac{\mathcal{M}_{sh}(\mathcal{D}'_m)(\boldsymbol{h})}{\mathcal{M}_{sh}(\mathcal{D}_m)(\boldsymbol{h})} - 1 \right|^\alpha \right] + \mathbb{E}_{\boldsymbol{h} \sim \mathcal{M}_{sh}(\mathcal{D}_m)} \left[ \left| \frac{\mathcal{M}_{sh}(\mathcal{D}''_m)(\boldsymbol{h})}{\mathcal{M}_{sh}(\mathcal{D}_m)(\boldsymbol{h})} - 1 \right|^\alpha \right] \right).
$$
$$(30)$$

*Proof.* Fix any $\boldsymbol{h} \in \mathcal{A}_B^m$, and take $x = \left( \frac{\mathcal{M}_{sh}(\mathcal{D}'_m)(\boldsymbol{h})}{\mathcal{M}_{sh}(\mathcal{D}_m)(\boldsymbol{h})} - 1 \right)$, $y = -\left( \frac{\mathcal{M}_{sh}(\mathcal{D}''_m)(\boldsymbol{h})}{\mathcal{M}_{sh}(\mathcal{D}_m)(\boldsymbol{h})} - 1 \right)$. Then applying Lemma 6 and taking expectation w.r.t. $\boldsymbol{h} \sim \mathcal{M}_{sh}(\mathcal{D}_m)$ will yield Corollary 1. ∎

**Remark 3.** Observe that the proof of Corollary 1 does not require $\mathcal{D}_m, \mathcal{D}'_m, \mathcal{D}''_m$ to be special triple of neighboring datasets such that $(\mathcal{D}_m, \mathcal{D}'_m, \mathcal{D}''_m) \in \mathcal{D}_{\text{same}}^m$. In fact, Corollary 1 holds for any triple of distributions $p, q, r$ over the same domain, for which we can show $\mathbb{E}_r \left[ \left| \frac{p-q}{r} \right|^\alpha \right] \leq 2^{\alpha-1} \left( \mathbb{E}_r \left[ \left| \frac{p}{r} - 1 \right|^\alpha \right] + \mathbb{E}_r \left[ \left| \frac{q}{r} - 1 \right|^\alpha \right] \right)$.

Now, in order to prove Theorem 6, it suffices to bound the expectation terms on the RHS of (30). This is what we do in the lemma below.

**Lemma 7.** *For any pair of the special pair of neighboring datasets $\mathcal{D}_m, \mathcal{D}'_m$, where $\mathcal{D}_m = (d, \ldots, d) \in \mathcal{X}^m$ and $\mathcal{D}'_m = (d, \ldots, d, d') \in \mathcal{X}^m$, we have*

$$
\mathbb{E}_{\boldsymbol{h} \sim \mathcal{M}_{sh}(\mathcal{D}_m)} \left[ \left| \frac{\mathcal{M}_{sh}(\mathcal{D}'_m)(\boldsymbol{h})}{\mathcal{M}_{sh}(\mathcal{D}_m)(\boldsymbol{h})} - 1 \right|^\alpha \right] \leq \begin{cases} \frac{(e^{\epsilon_0}-1)^2}{m e^{\epsilon_0}} & \text{if } \alpha = 2, \\ \alpha \Gamma(\alpha/2) \left( \frac{(e^{2\epsilon_0}-1)^2}{2m e^{2\epsilon_0}} \right)^{\alpha/2} & \text{otherwise.} \end{cases}
$$

Substituting the bound from Lemma 7 into Corollary 1, we get

$$
\mathbb{E}_{\boldsymbol{h} \sim \mathcal{M}_{sh}(\mathcal{D}_m)} \left[ \left| \frac{\mathcal{M}_{sh}(\mathcal{D}'_m)(\boldsymbol{h}) - \mathcal{M}_{sh}(\mathcal{D}''_m)(\boldsymbol{h})}{\mathcal{M}_{sh}(\mathcal{D}_m)(\boldsymbol{h})} \right|^\alpha \right] \leq \begin{cases} 4 \frac{(e^{\epsilon_0}-1)^2}{m e^{\epsilon_0}} & \text{if } \alpha = 2, \\ \alpha \Gamma(\alpha/2) \left( \frac{2(e^{2\epsilon_0}-1)^2}{m e^{2\epsilon_0}} \right)^{\alpha/2} & \text{otherwise,} \end{cases}
$$

which completes the proof of Theorem 6.

*Proof of Lemma 7.* Consider any pair of neighboring datasets $\mathcal{D}_m, \mathcal{D}'_m$, where $\mathcal{D}_m = (d, \ldots, d) \in \mathcal{X}^m$, $\mathcal{D}'_m = (d, \ldots, d, d') \in \mathcal{X}^m$. Let $\boldsymbol{p} = (p_1, \ldots, p_B)$ and $\boldsymbol{p}' = (p'_1, \ldots, p'_B)$ be the probability distributions of the discrete $\epsilon_0$-LDP mechanism $\mathcal{R} : \mathcal{X} \to \mathcal{Y} = [B]$ when its inputs are $d$ and $d'$ respectively, where $p_j = \Pr[\mathcal{R}(d) = j]$ and $p'_j = \Pr[\mathcal{R}(d') = j]$ for all $j \in [B]$. Since $\mathcal{R}$ is $\epsilon_0$-LDP, we have

$$
e^{-\epsilon_0} \leq \frac{p_j}{p'_j} \leq e^{\epsilon_0}, \qquad \forall j \in [B]. \tag{31}
$$

Since $\mathcal{M}_{sh}$ is a shuffled mechanism, it induces a distribution on $\mathcal{A}_B^m$ for any input dataset. So, for any $\boldsymbol{h} \in \mathcal{A}_B^m$, $\mathcal{M}_{sh}(\mathcal{D}_m)(\boldsymbol{h})$ and $\mathcal{M}_{sh}(\mathcal{D}'_m)(\boldsymbol{h})$ are equal to the probabilities of seeing $\boldsymbol{h}$ when the inputs to $\mathcal{M}_{sh}$ are $\mathcal{D}_m$ and $\mathcal{D}'_m$, respectively. Thus, for a given histogram $\boldsymbol{h} = (h_1, \ldots, h_B) \in \mathcal{A}_B^m$ with $m$ elements and $B$ bins, we have

$$
\mathcal{M}_{sh}(\mathcal{D}_m)(\boldsymbol{h}) = MN(m, \boldsymbol{p}, \boldsymbol{h}) = \binom{m}{\boldsymbol{h}} \prod_{j=1}^{B} p_j^{h_j}, \tag{32}
$$

$$
\mathcal{M}_{sh}(\mathcal{D}'_m)(\boldsymbol{h}) = \sum_{j=1}^{B} p'_j MN\left(m-1, \boldsymbol{p}, \widetilde{\boldsymbol{h}}_j\right), \tag{33}
$$

where $MN(m, \boldsymbol{p}, \boldsymbol{h})$ denotes the Multinomial distribution with $\binom{m}{\boldsymbol{h}} = \frac{m!}{h_1! \cdots h_B!}$.

Let $X : \mathcal{A}_B^m \to \mathbb{R}$ be a random variable associated with the distribution $\mathcal{M}(\mathcal{D}_m)$ on $\mathcal{A}_B^m$, and for any $\boldsymbol{h} \in \mathcal{A}_B^m$, define $X(\boldsymbol{h}) := m\left(\frac{\mathcal{M}(\mathcal{D}'_m)(\boldsymbol{h})}{\mathcal{M}(\mathcal{D}_m)(\boldsymbol{h})} - 1\right)$. Thus, we get that $\mathbb{E}\left[\left|\frac{\mathcal{M}_{sh}(\mathcal{D}'_m)(\boldsymbol{h})}{\mathcal{M}_{sh}(\mathcal{D}_m)(\boldsymbol{h})} - 1\right|^\alpha\right] = \frac{1}{m^\alpha}\mathbb{E}\left[|X|^\alpha\right]$. From [29, Lemmas 6, 7], we get that

$$\frac{1}{m^\alpha}\mathbb{E}\left[|X|^\alpha\right] \leq \begin{cases} \frac{(e^{\epsilon_0}-1)^2}{me^{\epsilon_0}} & \text{if } \alpha = 2, \\ \alpha\Gamma\left(\alpha/2\right)\left(\frac{\left(e^{2\epsilon_0}-1\right)^2}{2me^{2\epsilon_0}}\right)^{\alpha/2} & \text{otherwise.} \end{cases} \tag{34}$$

This completes the proof of Lemma 7. ∎

## C.4 Completing the Proof of Theorem 4

For simplicity of notation, for any $m \in \{0, 1, \ldots, n-1\}$, define

$$q_m := \binom{k-1}{m}q^m(1-q)^{k-m-1}$$

$$E_m := \mathbb{E}_{\boldsymbol{h}\sim\mathcal{M}_{sh}(\mathcal{D}''^{(k)}_{m+1})}\left[\left|\frac{\mathcal{M}_{sh}(\mathcal{D}^{(k)}_{m+1})(\boldsymbol{h}) - \mathcal{M}_{sh}(\mathcal{D}'^{(k)}_{m+1})(\boldsymbol{h})}{\mathcal{M}_{sh}(\mathcal{D}''^{(k)}_{m+1})(\boldsymbol{h})}\right|^\alpha\right].$$

First we show an important property of $E_m$ that we will use in the proof.

**Lemma 8.** $E_m$ is a non-increasing function of $m$, i.e.,

$$\mathbb{E}_{\boldsymbol{h}\sim\mathcal{M}_{sh}(\mathcal{D}''^{(k)}_{m+1})}\left[\left|\frac{\mathcal{M}_{sh}(\mathcal{D}^{(k)}_{m+1})(\boldsymbol{h}) - \mathcal{M}_{sh}(\mathcal{D}''^{(k)}_{m+1})(\boldsymbol{h})}{\mathcal{M}_{sh}(\mathcal{D}''^{(k)}_{m+1})(\boldsymbol{h})}\right|^\alpha\right]$$

$$\leq \mathbb{E}_{\boldsymbol{h}\sim\mathcal{M}_{sh}(\mathcal{D}''^{(k)}_{m})}\left[\left|\frac{\mathcal{M}_{sh}(\mathcal{D}^{(k)}_{m})(\boldsymbol{h}) - \mathcal{M}_{sh}(\mathcal{D}'^{(k)}_{m})(\boldsymbol{h})}{\mathcal{M}_{sh}(\mathcal{D}''^{(k)}_{m})(\boldsymbol{h})}\right|^\alpha\right], \quad (35)$$

where, for any $l \in \{m, m+1\}$, $\mathcal{D}^{(k)}_l = (d''_k, \ldots, d''_k, d_k)$, $\mathcal{D}'^{(k)}_l = (d''_k, \ldots, d''_k, d'_k)$, and $\mathcal{D}''^{(k)}_l = (d''_k, \ldots, d''_k, d''_k)$, each having $l$ elements.

*Proof.* Lemma 8 follows from Lemma 5 in a straightforward manner, as Lemma 5 is for arbitrary triples of adjacent datasets, whereas, Lemma 8 is for triples of adjacent datasets having special structures. ∎

Thus, we get

$$\mathbb{E}_{\boldsymbol{h}\sim\mathcal{M}_{sh}(\mathcal{D}'')}\left[\left|\frac{\mathcal{M}_{sh}(\mathcal{D})(\boldsymbol{h}) - \mathcal{M}_{sh}(\mathcal{D}')(\boldsymbol{h})}{\mathcal{M}_{sh}(\mathcal{D}'')(\boldsymbol{h})}\right|^\alpha\right]$$

$$\leq \sum_{m=0}^{k-1} q_m E_m$$

$$= \sum_{m<\lfloor(1-\gamma)q(k-1)\rfloor} q_m E_m + \sum_{m\geq\lfloor(1-\gamma)q(k-1)\rfloor} q_m E_m$$

$$\overset{(a)}{\leq} E_0 \sum_{m<\lfloor(1-\gamma)q(k-1)\rfloor} q_m + \sum_{m\geq\lfloor(1-\gamma)q(k-1)\rfloor} q_m E_m$$

$$\overset{(b)}{\leq} E_0 e^{-\frac{q(k-1)\gamma^2}{2}} + \sum_{m\geq\lfloor(1-\gamma)q(k-1)\rfloor} q_m E_m$$

$$\overset{(c)}{\leq} e^{\epsilon_0\alpha} e^{-\frac{q(k-1)\gamma^2}{2}} + \sum_{m\geq\lfloor(1-\gamma)q(k-1)\rfloor} q_m E_m$$

$$\overset{(d)}{\leq} (e^{\epsilon_0} - e^{-\epsilon_0})^\alpha e^{-\frac{q(k-1)\gamma^2}{2}} + E_{(1-\gamma)q(k-1)}. \tag{36}$$

Here, steps (a) and (d) follow from the fact that $E_m$ is a non-increasing function of $m$ (see Lemma 8). Step (b) follows from the Chernoff bound. In step (c), we used that $\mathcal{M}_{sh}(d_k) = \mathcal{R}(d_k)$, $\mathcal{M}_{sh}(d'_k) = \mathcal{R}(d'_k)$, and $\mathcal{M}_{sh}(d''_k) = \mathcal{R}(d''_k)$ which together imply that $E_0 = (e^{\epsilon_0} - e^{-\epsilon_0})^\alpha$, where the inequality follows because $\mathcal{R}$ is an $\epsilon_0$-LDP mechanism. By choosing $\gamma = 0.5$ completes the proof of Theorem 4.

## D Proof of Theorem 2 (Lower Bound)

Consider the binary case, where each data point $d$ can take a value from $\mathcal{X} = \{0, 1\}$. Let the local randomizer $\mathcal{R}$ be the binary randomized response (2RR) mechanism, where $\Pr[\mathcal{R}(d) = d] = \frac{e^{\epsilon_0}}{e^{\epsilon_0}+1}$ for $d \in \mathcal{X}$. It is easy to verify that $\mathcal{R}$ is an $\epsilon_0$-LDP mechanism. For simplicity, let $p = \frac{1}{e^{\epsilon_0}+1}$. Consider two neighboring datasets $\mathcal{D}, \mathcal{D}' \in \{0, 1\}^k$, where $\mathcal{D} = (0, \dots, 0, 0)$ and $\mathcal{D}' = (0, \dots, 0, 1)$. Let $m \in \{0, \dots, k\}$ denote the number of ones in the output of the shuffler. We define two distributions

$$\mu_0(m) = \binom{k}{m} p^m (1-p)^{k-m},$$

$$\mu_1(m) = (1-p)\binom{k-1}{m-1}p^{m-1}(1-p)^{k-m} + p\binom{k-1}{m}p^m(1-p)^{k-m-1}.$$

(37)

As argued on page 5, since the output of the shuffled mechanism $\mathcal{M}$ can be thought of as the distribution of the number of ones in the output, we have that $m \sim \mathcal{M}(\mathcal{D})$ is distributed as a Binomial random variable $\text{Bin}(k, p)$. Thus, we have

$$\mathcal{M}(\mathcal{D})(m) = \mu_0(m)$$
$$\mathcal{M}(\mathcal{D}')(m) = (1-\gamma)\mu_1(m) + \gamma\mu_0(m)$$

It will be useful to compute $\frac{\mu_1(m)}{\mu_0(m)} - 1$ for the calculations later.

$$\frac{\mu_1(m)}{\mu_0(m)} - 1 = \frac{(1-p)\binom{k-1}{m-1}p^{m-1}(1-p)^{k-m} + p\binom{k-1}{m}p^m(1-p)^{k-m-1}}{\binom{k}{m}p^m(1-p)^{k-m}} - 1$$

$$= \frac{m}{k}\frac{(1-p)}{p} + \frac{(k-m)}{k}\frac{p}{(1-p)} - 1$$

$$= \frac{m}{k}e^{\epsilon_0} + \frac{(k-m)}{k}e^{-\epsilon_0} - 1$$

$$= \frac{m}{k}\left(e^{\epsilon_0} - e^{-\epsilon_0}\right) + e^{-\epsilon_0} - 1$$

$$= \frac{m}{k}\left(\frac{e^{2\epsilon_0} - 1}{e^{\epsilon_0}}\right) - \left(\frac{e^{\epsilon_0} - 1}{e^{\epsilon_0}}\right)$$

$$= \left(\frac{e^{2\epsilon_0} - 1}{ke^{\epsilon_0}}\right)\left(m - \frac{k}{e^{\epsilon_0}+1}\right)$$

(38)

Thus, we have that

$$\mathbb{E}_{m\sim\mathcal{M}(\mathcal{D})}\left[\left(\frac{\mathcal{M}(\mathcal{D}')(m)}{\mathcal{M}(\mathcal{D})(m)}\right)^\lambda\right] = \mathbb{E}\left[\left(1 + \gamma\left(\frac{\mu_1(m)}{\mu_0(m)} - 1\right)\right)^\lambda\right]$$

$$\overset{(a)}{=} 1 + \sum_{i=1}^{\lambda}\binom{\lambda}{i}\gamma^i\mathbb{E}\left[\left(\frac{\mu_1(m)}{\mu_0(m)} - 1\right)^i\right]$$

$$\overset{(b)}{=} 1 + \sum_{i=2}^{\lambda}\binom{\lambda}{i}\gamma^i\mathbb{E}\left[\left(\frac{\mu_1(m)}{\mu_0(m)} - 1\right)^i\right]$$

$$= 1 + \sum_{i=2}^{\lambda}\binom{\lambda}{i}\gamma^i\left(\frac{(e^{2\epsilon_0} - 1)}{ke^{\epsilon_0}}\right)^i\mathbb{E}\left[\left(m - \frac{k}{e^{\epsilon_0}+1}\right)^i\right] \qquad \text{(from (38))}$$

$$\overset{(c)}{=} 1 + \binom{\lambda}{2}\gamma^2\frac{(e^{\epsilon_0} - 1)^2}{ke^{\epsilon_0}} + \sum_{i=3}^{\lambda}\binom{\lambda}{i}\gamma^i\left(\frac{(e^{2\epsilon_0} - 1)}{ke^{\epsilon_0}}\right)^i\mathbb{E}\left[\left(m - \frac{k}{e^{\epsilon_0}+1}\right)^i\right].$$

Here, step (a) from the polynomial expansion $(1 + x)^k = \sum_{m=0}^{k} \binom{k}{m} x^m$, step (b) follows because the term corresponding to $i = 1$ is zero (i.e., $\mathbb{E}_{m \sim \mu_0} \left[ \left( \frac{\mu_1(m)}{\mu_0(m)} - 1 \right) \right] = 0$), and step (c) from the from the fact that $\mathbb{E}_{m \sim \mu_0} \left[ \left( m - \frac{k}{e^{\epsilon_0} + 1} \right)^2 \right] = kp(1 - p) = \frac{k e^{\epsilon_0}}{(e^{\epsilon_0} + 1)^2}$, which is equal to the variance of the Binomial random variable. This completes the proof of Theorem 2.

# E  Omitted Details from Section 5

## E.1  Proof of Lemma 2

Consider arbitrary neighboring datasets $\mathcal{D} = (d_1, \ldots, d_n) \in \mathcal{X}^n$ and $\mathcal{D}' = (d_1, \ldots, d_{n-1}, d'_n) \in \mathcal{X}^n$. Recall that the LDP mechanism $\mathcal{R} : \mathcal{X} \to \mathcal{Y}$ has a discrete range $\mathcal{Y} = [B]$ for some $B \in \mathbb{N}$. Let $\boldsymbol{p}_i := (p_{i1}, \ldots, p_{iB})$ and $\boldsymbol{p}'_n := (p'_{n1}, \ldots, p'_{nB})$ denote the probability distributions over $\mathcal{Y}$ when the input to $\mathcal{R}$ is $d_i$ and $d'_n$, respectively, where $p_{ij} = \Pr[\mathcal{R}(d_i) = j]$ and $p'_{nj} = \Pr[\mathcal{R}(d'_n) = j]$ for all $j \in [B]$ and $i \in [n]$.

Let $\mathcal{P} = \{\boldsymbol{p}_i : i \in [n]\}$ and $\mathcal{P}' = \{\boldsymbol{p}_i : i \in [n-1]\} \bigcup \{\boldsymbol{p}'_n\}$. For $i \in [n-1]$, let $\mathcal{P}_{-i} = \mathcal{P} \setminus \{\boldsymbol{p}_i\}$, $\mathcal{P}'_{-i} = \mathcal{P}' \setminus \{\boldsymbol{p}_i\}$, and also $\mathcal{P}_{-n} = \mathcal{P} \setminus \{\boldsymbol{p}_n\}$, $\mathcal{P}'_{-n} = \mathcal{P}' \setminus \{\boldsymbol{p}'_n\}$.

Here, $\mathcal{P}, \mathcal{P}'$ correspond to the datasets $\mathcal{D} = \{d_1, \ldots, d_n\}, \mathcal{D}' = \{d_1, \ldots, d_{n-1}, d'_n\}$, respectively, and for any $i \in [n]$, $\mathcal{P}_{-i}$ and $\mathcal{P}'_{-i}$ correspond to the datasets $\mathcal{D}_{-i} = \{d_1, \ldots, d_{i-1}, d_{i+1}, \ldots, d_n\}$ and $\mathcal{D}'_{-i} = \{d_1, \ldots, d_{i-1}, d_{i+1}, \ldots, d_{n-1}, d'_n\}$, respectively. Thus, without loss of generality, we deal with sets $\mathcal{P}$ and $\mathcal{P}'$ throughout this section instead of dealing with $\mathcal{D}$ and $\mathcal{D}'$. Thus, we write $\mathcal{M}(\mathcal{P}) \triangleq \mathcal{M}(\mathcal{D})$ and $\mathcal{M}(\mathcal{P}') \triangleq \mathcal{M}(\mathcal{D}')$.

We bound the Rényi divergence between $\mathcal{M}(\mathcal{P})$ and $\mathcal{M}(\mathcal{P}')$. For given a set $\mathcal{S} \subset [n]$ with $|\mathcal{S}| = \gamma n = k$, we define two sets $\mathcal{P}^{\mathcal{S}}, \mathcal{P}'^{\mathcal{S}}$, having $k$ distributions each, as follows:

$$\mathcal{P}^{\mathcal{S}} = \{\boldsymbol{p}_i : i \in \mathcal{S}\}, \tag{39}$$

$$\mathcal{P}'^{\mathcal{S}} = \{\boldsymbol{p}_i : i \in \mathcal{S}\}, \tag{40}$$

Observe that when $n \notin \mathcal{S}$, we have that $\mathcal{P}^{\mathcal{S}} = \mathcal{P}'^{\mathcal{S}}$. For given $k$ distributions $\mathcal{P} = (\boldsymbol{p}_1, \ldots, \boldsymbol{p}_k)$, we define a shuffle mechanism $\mathcal{M}_{sh}(\mathcal{P})$ as follows:

$$\mathcal{M}_{sh}(\mathcal{P}) = \mathcal{H}_k(\boldsymbol{p}_1, \ldots, \boldsymbol{p}_k). \tag{41}$$

Thus, the mechanisms $\mathcal{M}(\mathcal{P})$ and $\mathcal{M}(\mathcal{P}')$ can be defined by:

$$\mathcal{M}(\mathcal{P}) = \frac{1}{\binom{n}{k}} \sum_{\mathcal{S} \subset [n]} \mathcal{M}_{sh}(\mathcal{P}^{\mathcal{S}})$$

$$= \gamma P_E + (1 - \gamma) P_{E^c} \tag{42}$$

$$\mathcal{M}(\mathcal{P}') = \frac{1}{\binom{n}{k}} \sum_{\mathcal{S} \subset [n]} \mathcal{M}_{sh}(\mathcal{P}'^{\mathcal{S}})$$

$$= \gamma Q_E + (1 - \gamma) P_{E^c}, \tag{43}$$

where $P_E = \frac{1}{\binom{n-1}{k-1}} \sum_{\substack{\mathcal{S} \subset [n] \\ n \in \mathcal{S}}} \mathcal{M}_{sh}(\mathcal{P}^{\mathcal{S}})$, $P_{E^c} = \frac{1}{\binom{n-1}{k}} \sum_{\substack{\mathcal{S} \subset [n] \\ n \notin \mathcal{S}}} \mathcal{M}_{sh}(\mathcal{P}^{\mathcal{S}})$, and $Q_E = \frac{1}{\binom{n-1}{k-1}} \sum_{\substack{\mathcal{S} \subset [n] \\ n \in \mathcal{S}}} \mathcal{M}_{sh}(\mathcal{P}'^{\mathcal{S}})$. Hence, from the polynomial expansion, we get:

$$
\begin{aligned}
\mathbb{E}_{\boldsymbol{h} \sim \mathcal{M}(\mathcal{P}')} \left[ \left( \frac{\mathcal{M}(\mathcal{P})(\boldsymbol{h})}{\mathcal{M}(\mathcal{P}')(\boldsymbol{h})} \right)^{\lambda} \right] &= \mathbb{E}_{\boldsymbol{h} \sim \mathcal{M}(\mathcal{P}')} \left[ \left( 1 + \frac{\mathcal{M}(\mathcal{P})(\boldsymbol{h})}{\mathcal{M}(\mathcal{P}')(\boldsymbol{h})} - 1 \right)^{\lambda} \right] \\
&= 1 + \sum_{j=2}^{\lambda} \binom{\lambda}{j} \mathbb{E}_{\boldsymbol{h} \sim \mathcal{M}(\mathcal{P}')} \left[ \left( \frac{\mathcal{M}(\mathcal{P})(\boldsymbol{h}) - \mathcal{M}(\mathcal{P}')(\boldsymbol{h})}{\mathcal{M}(\mathcal{P}')(\boldsymbol{h})} \right)^{j} \right] \\
&= 1 + \sum_{j=2}^{\lambda} \binom{\lambda}{j} \gamma^j \mathbb{E}_{\boldsymbol{h} \sim \mathcal{M}(\mathcal{P}')} \left[ \left( \frac{P_E(\boldsymbol{h}) - Q_E(\boldsymbol{h})}{\mathcal{M}(\mathcal{P}')(\boldsymbol{h})} \right)^{j} \right]
\end{aligned}
$$

$$\tag{44}$$

Now, we borrow the trick used in [43] to bound each term in the right hand side in (44). For completeness, we repeat their definitions and proofs here. We define an auxiliary dummy variable $i \sim \text{Unif}(1, \ldots, k)$ that is independent to everything else. Furthermore, we define two functions $g(\mathcal{S}, i)$ and $g'(\mathcal{S}, i)$ as follows:

$$g(\mathcal{S}, i) = \begin{cases} \mathcal{M}_{sh}\left(\mathcal{P}^{\mathcal{S}}\right) & \text{if } n \in \mathcal{S} \\ \mathcal{M}_{sh}\left(\mathcal{P}^{\mathcal{S} \cup \{n\} \backslash \mathcal{S}(i)}\right) & \text{otherwise} \end{cases} \tag{45}$$

$$g'(\mathcal{S}, i) = \begin{cases} \mathcal{M}_{sh}\left(\mathcal{P}'^{\mathcal{S}}\right) & \text{if } n \in \mathcal{S} \\ \mathcal{M}_{sh}\left(\mathcal{P}'^{\mathcal{S} \cup \{n\} \backslash \mathcal{S}(i)}\right) & \text{otherwise} \end{cases} \tag{46}$$

Observe that $\mathbb{E}_{\mathcal{S},i}[g(\mathcal{S}, i)] = P_E$, $\mathbb{E}_{\mathcal{S},i}[g'(\mathcal{S}, i)] = Q_E$, and $\mathbb{E}_{\mathcal{S},i}\left[\mathcal{M}_{sh}\left(\mathcal{P}'^{\mathcal{S}}\right)\right] = \mathcal{M}(\mathcal{P})$. As a result, we get that

$$\mathbb{E}_{\boldsymbol{h} \sim \mathcal{M}(\mathcal{P}')}\left[\left(\frac{P_E(\boldsymbol{h}) - Q_E(\boldsymbol{h})}{\mathcal{M}(\mathcal{P}')(\boldsymbol{h})}\right)^j\right] \leq \sum_{\boldsymbol{h}} \frac{|P_E(\boldsymbol{h}) - Q_E(\boldsymbol{h})|^j}{(\mathcal{M}(\mathcal{P}')(\boldsymbol{h}))^{j-1}}$$

$$\leq \sum_{\boldsymbol{h}} \mathbb{E}_{\mathcal{S},i}\left[\frac{|g(\mathcal{S}, i)(\boldsymbol{h}) - g'(\mathcal{S}, i)(\boldsymbol{h})|^j}{(\mathcal{M}_{sh}(\mathcal{P}'^{\mathcal{S}})(\boldsymbol{h}))^{j-1}}\right]$$

$$= \mathbb{E}_{\mathcal{S},i}\mathbb{E}_{\boldsymbol{h} \sim \mathcal{M}_{sh}(\mathcal{P}'^{\mathcal{S}})}\left[\left(\frac{|g(\mathcal{S}, i)(\boldsymbol{h}) - g'(\mathcal{S}, i)(\boldsymbol{h})|}{\mathcal{M}_{sh}(\mathcal{P}'^{\mathcal{S}})(\boldsymbol{h})}\right)^j\right]$$

$$\leq (\zeta(j))^j.$$

Here, step (a) follows from Jensen's inequality and the convexity of the function $x^j/y^{j-1}$ (see [43, Lemma 20]). Step (b) follows from Fubini's theorem. The last inequality is obtained by taking the supremum over all possible neighboring datasets $\mathcal{D}, \mathcal{D}'$. This completes the proof of Lemma 2.

### E.2   Combining Theorem 4 and Lemma 2 for Proving Theorem 1

In this section, we complete the remaining calculation from Section 5 of combining the results of Theorem 4 and Lemma 2 and proving Theorem 1.

Substituting the bound on $\zeta(\alpha)$ from Theorem 4 into Lemma 2, we get

$$\epsilon(\lambda) \leq \frac{1}{\lambda - 1} \log\left[1 + \binom{\lambda}{2}\gamma^2\left(4\frac{(e^{\epsilon_0} - 1)^2}{\bar{k}e^{\epsilon_0}} + (e^{\epsilon_0} - e^{-\epsilon_0})^2 e^{-\frac{k-1}{8e^{\epsilon_0}}}\right)\right.$$

$$\left. + \sum_{\alpha=3}^{\lambda}\binom{\lambda}{\alpha}\gamma^\alpha\left(\alpha\Gamma(\alpha/2)\left(\frac{2(e^{2\epsilon_0} - 1)^2}{\bar{k}e^{2\epsilon_0}}\right)^{\alpha/2} + (e^{\epsilon_0} - e^{-\epsilon_0})^\alpha e^{-\frac{k-1}{8e^{\epsilon_0}}}\right)\right]$$

$$= \frac{1}{\lambda - 1}\log\left[1 + 4\binom{\lambda}{2}\gamma^2\frac{(e^{\epsilon_0} - 1)^2}{\bar{k}e^{\epsilon_0}} + \sum_{\alpha=3}^{\lambda}\binom{\lambda}{\alpha}\gamma^\alpha\alpha\Gamma(\alpha/2)\left(\frac{2(e^{2\epsilon_0} - 1)^2}{\bar{k}e^{2\epsilon_0}}\right)^{\alpha/2} + \Upsilon\right], \tag{47}$$

where $\Upsilon = \sum_{\alpha=2}^{\lambda}\binom{\lambda}{\alpha}\gamma^\alpha(e^{\epsilon_0} - e^{-\epsilon_0})^\alpha e^{-\frac{k-1}{8e^{\epsilon_0}}} = \left(\left(1 + \gamma\frac{e^{2\epsilon_0} - 1}{e^{\epsilon_0}}\right)^\lambda - 1 - \lambda\gamma\frac{e^{2\epsilon_0} - 1}{e^{\epsilon_0}}\right)e^{-\frac{k-1}{8e^{\epsilon_0}}}$.

The above expression in (47) is the bound given in Theorem 1.

## F   Proof of Theorem 3: Privacy-Convergence Tradeoff

In this section, we prove the privacy-convergence tradeoff of Algorithm 1 and prove Theorem 3.

The privacy part is straightforward from conversion from RDP to approximate DP using Lemma 1 and Theorem 1. Now, we prove the convergence rate.

At iteration $t \in [T]$ of Algorithm 1, server averages the $k$ received gradients and obtains $\overline{\mathbf{g}}_t = \frac{1}{k}\sum_{i \in \mathcal{U}_t} \mathcal{R}_p(\tilde{\mathbf{g}}_t(d_i))$ and then updates the parameter vector as $\theta_{t+1} \leftarrow \prod_{\mathcal{C}}(\theta_t - \eta_t\overline{\mathbf{g}}_t)$. Observe that the mechanism $\mathcal{R}_p$ is unbiased and has a bounded variance: $\sup_{\mathbf{x} \in \mathcal{B}_p(L)} \mathbb{E}\|\mathcal{R}_p(\mathbf{x}) - \mathbf{x}\|_2^2 \leq G_p^2(L)$. As a result, the average gradient $\overline{\mathbf{g}}_t$ is also unbiased, i.e., we have $\mathbb{E}[\overline{\mathbf{g}}_t] = \nabla_{\theta_t} F(\theta_t)$, where expectation is taken with respect to the random subsampling of clients as well as the randomness of the mechanism $\mathcal{R}_p$. Now we show that $\overline{\mathbf{g}}_t$ has a bounded second moment.

**Lemma 9.** *For any $d \in \mathcal{X}$, if the function $f(\theta; .) : \mathcal{C} \times \mathcal{X} \to \mathbb{R}$ is convex and $L$-Lipschitz continuous with respect to the $\ell_g$-norm, which is the dual of the $\ell_p$-norm (i.e., $\frac{1}{p} + \frac{1}{g} = 1$), then we have*

$$\mathbb{E}\|\overline{\mathbf{g}}_t\|_2^2 \leq L^2 \max\{d^{1-\frac{2}{p}}, 1\}\left(1 + \frac{cd}{qn}\left(\frac{e^{\epsilon_0} + 1}{e^{\epsilon_0} - 1}\right)^2\right), \tag{48}$$

*where $c$ is a global constant: $c = 4$ if $p \in \{1, \infty\}$ and $c = 14$ if $p \notin \{1, \infty\}$.*

*Proof.* Under the conditions of the lemma, we have from [40, Lemma 2.6] that $\|\nabla_\theta f(\theta; d)\| \leq L$ for all $d \in \mathcal{X}$, which implies that $\|\nabla_\theta F(\theta)\| \leq L$. Thus, we have

$$\mathbb{E}\|\overline{\mathbf{g}}_t\|_2^2 = \|\mathbb{E}[\overline{\mathbf{g}}_t]\|_2^2 + \mathbb{E}\|\overline{\mathbf{g}}_t - \mathbb{E}[\overline{\mathbf{g}}_t]\|_2^2$$

$$\overset{(a)}{\leq} \max\{d^{1-\frac{2}{p}}, 1\}L^2 + \mathbb{E}\|\overline{\mathbf{g}}_t - \mathbb{E}[\overline{\mathbf{g}}_t]\|_2^2$$

$$\overset{(b)}{\leq} \max\{d^{1-\frac{2}{p}}, 1\}L^2 + \frac{G_p(L)^2}{k}$$

$$\overset{(c)}{=} \max\{d^{1-\frac{2}{p}}, 1\}L^2 + \frac{G_p^2(L)}{\gamma n},$$

Step $(a)$ follows from the fact that $\|\nabla_{\theta_t} F(\theta_t)\| \leq L$ together with the norm inequality $\|\mathbf{u}\|_q \leq \|\mathbf{u}\|_p \leq d^{\frac{1}{p} - \frac{1}{q}}\|\mathbf{u}\|_q$ for $1 \leq p \leq q \leq \infty$. Step $(b)$ follows from the assumption that $\mathcal{R}_p$ has bounded variance. Step $(c)$ uses $\gamma = \frac{k}{n}$. ∎

Now, we can use standard SGD convergence results for convex functions. In particular, we use the following result from [41].

**Lemma 10** (SGD Convergence [41]). *Let $F(\theta)$ be a convex function, and the set $\mathcal{C}$ has diameter $D$. Consider a stochastic gradient descent algorithm $\theta_{t+1} \leftarrow \prod_{\mathcal{C}}(\theta_t - \eta_t\mathbf{g}_t)$, where $\mathbf{g}_t$ satisfies $\mathbb{E}[\mathbf{g}_t] = \nabla_{\theta_t} F(\theta_t)$ and $\mathbb{E}\|\mathbf{g}_t\|_2^2 \leq G^2$. By setting $\eta_t = \frac{D}{G\sqrt{t}}$, we get*

$$\mathbb{E}[F(\theta_T)] - F(\theta^*) \leq 2DG\frac{2 + \log(T)}{\sqrt{T}} = \mathcal{O}\left(DG\frac{\log(T)}{\sqrt{T}}\right). \tag{49}$$

As shown in Lemma 9 and above that Algorithm 1 satisfies the premise of Lemma 10. Now, using the bound on $G^2$ from Lemma 9, we have that the output $\theta_T$ of Algorithm 1 satisfies

$$\mathbb{E}[F(\theta_T)] - F(\theta^*) \leq \mathcal{O}\left(DG\frac{\log(T)}{\sqrt{T}}\right), \tag{50}$$

where $G^2 = \max\{d^{1-\frac{2}{p}}, 1\}L^2 + \frac{G_p^2(L)}{\gamma n}$. This completes the proof of the third part of Theorem 3.

# G Additional Numerical Results

In this section, we present additional numerical experiments of our bounds on RDP of the subsampled shuffle mechanism and its usage for getting approximate DP of Algorithm 1 for training machine learning models.

**Composition of a sequence of subsampled shuffle models:** Observe that our RDP bound presented in 1 is general for any values of LDP parameter $\epsilon_0$, number of clients $n$, and RDP order $\lambda \geq 2$. On the other hand, the results of the privacy amplification by shuffling presented in [24] is valid under the condition on the LDP parameter:

$$\epsilon_0 \leq \log\left(\frac{n}{16\log(2/\delta)}\right). \tag{51}$$

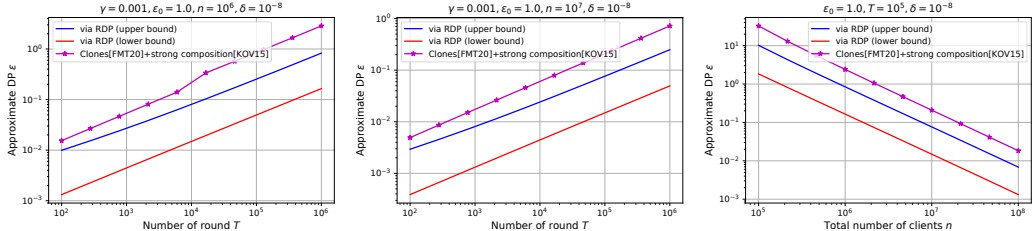

(a) Approx. DP as a function of $T$ for $\epsilon_0 = 1, \gamma = 0.001, n = 10^6$
(b) Approx. DP as a function of $T$ for $\epsilon_0 = 1, \gamma = 0.001, n = 10^7$
(c) Approx. DP as a function of $n$ for $\epsilon_0 = 1, \gamma n = 10^4, T = 10^5$

Figure 4: Comparison of our bound on the Approximate $(\epsilon, \delta)$-DP (blue) for composition of a sequence of subsampled shuffle mechanisms for $\delta = 10^{-8}$ with applying the strong composition theorem [34] after getting the Approximate DP of the shuffled model given in [24] with subsampling [42] (magenta).

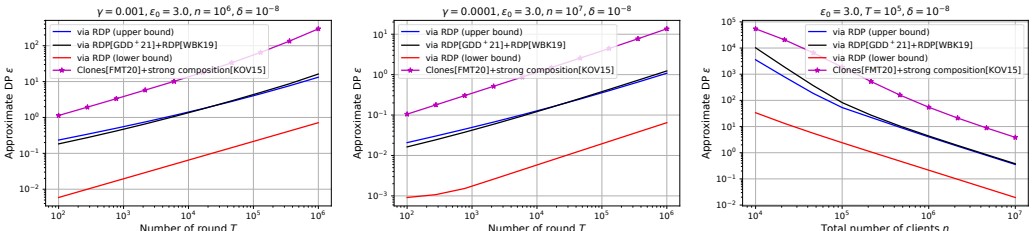

(a) Approx. DP as a function of $T$ for $\epsilon_0 = 3, \gamma = 0.001, n = 10^6$
(b) Approx. DP as a function of $T$ for $\epsilon_0 = 3, \gamma = 0.0001, n = 10^7$
(c) Approx. DP as a function of $n$ for $\epsilon_0 = 3, \gamma n = 10^3, T = 10^5$

Figure 5: Comparison of several bounds on the Approximate $(\epsilon, \delta)$-DP for composition of a sequence of subsampled shuffle mechanisms for $\delta = 10^{-8}$: (i) Approximate DP obtained from our upper bound on the RDP in Theorem 1 (blue); (ii) Approximate DP obtained from our lower bound on the RDP in Theorem 2 (red); (iii) Approximate DP obtained from the upper bound on the RDP given in [29] with RDP amplification by subsampling from [43] (black); and (iv) Applying the strong composition theorem [34] after getting the Approximate DP of the shuffled model given in [24] with subsampling [42] (magenta).

Furthermore, the results of the privacy amplification by shuffling presented in [7] is valid under the condition on the LDP parameter:

$$\epsilon_0 \le \frac{1}{2} \log\left(\frac{n}{\log(1/\delta)}\right). \tag{52}$$

Thus, if the conditions in (51)- (52) do not hold, then the results in [24, 7] have a privacy bound: $\epsilon = \epsilon_0$ and $\delta = 0$. For example, when total number of clients $n = 10^6$, LDP parameter $\epsilon_0 = 3$, and we choose uniformly at random $k = 1000$ clients at each iteration, then both conditions (51)- (52) do not hold.

In Figure 4, we plot additional results for comparison between our bound on the approximate DP for a composition of $T$ mechanisms $(\mathcal{M}_1, \ldots, \mathcal{M}_T)$ with the bound obtained by applying the strong composition theorem [34] after getting the approximate DP of the shuffled model given in [24] with subsampling [42]. In Figure 5, we plot our bound on the approximate $(\epsilon, \delta)$-DP for a composition of $T$ mechanisms $(\mathcal{M}_1, \ldots, \mathcal{M}_T)$, where $\mathcal{M}_t$ is a subsampled shuffle mechanism for $t \in [T]$. In all our experiments reported in Figure 5, we fix $\delta = 10^{-8}$ and $\epsilon_0 = 3$, where we consider the cases in which the conditions (51)- (52) do not hold. Thus, we compare our results with the bound: $\epsilon = \epsilon_0$ and $\delta = 0$.

We observe that our new bound on the RDP of the subsampled shuffle mechanism achieves a significant saving in total privacy $\epsilon$ compared to the bound in [24, 7]. For example, we save a factor of $17\times$ compared to the results in [24, 7] with the strong composition theorem [34] in computing the overall privacy parameter $\epsilon$ for number of iterations $T = 10^5$, subsampling parameter $\gamma = 0.001$,

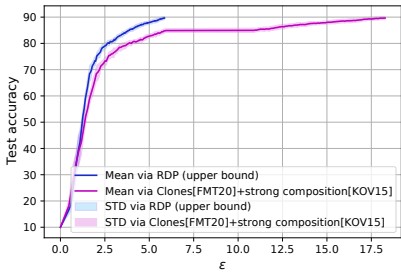

Figure 6: Privacy-Utility trade-offs on the MNIST dataset with $\ell_2$-norm clipping.

LDP parameter $\epsilon_0 = 3$, and number of clients $n = 10^6$. We observe that our bound on the over all privacy parameters almost matches the bound on the RDP given in [29] with subsampled RDP [43] when subsampling parameter $\gamma = 0.001$, LDP parameter $\epsilon_0 = 3$, and number of clients $n = 10^6$.

**Distributed private learning:** We numerically evaluate the proposed privacy-learning performance on training machine learning models. We train a simple neural network on MNIST dataset that was also used in [21, 39] and described in Table 1. We assume that we have $n = 60,000$ clients, where each client has one sample. At each step of the Algorithm 1, we choose uniformly at random $2,000$ clients, where each client clips the $\ell_2$-norm of the gradient with clipping parameter $C = 0.005$ and applies the $\mathcal{R}_2$ $\epsilon_0$-LDP mechanism (Privunit) proposed in [17] with $\epsilon_0 = 2$. We run Algorithm 1 with $\delta = 10^{-5}$ for 200 epochs, with learning rate $\eta = 12$ for the first 30 epochs, and then decrease it to 4 in the next 30 epochs. We decrease the learning rate to $3.5$ for the remaining epochs.

Figure 6 plots the mean and the standard deviation of privacy-accuracy trade-offs averaged over 4 runs. We observe that we achieve an accuracy of $81.15\%(\pm 0.7)$ with a total privacy budget of $\epsilon = 3$ using our new privacy analysis, whereas, [24] achieves an accuracy of only $76.46\%(\pm 1.9)$ with the same privacy budget of $\epsilon = 3$ using the standard composition theorems. Furthermore, we can see that we achieves accuracy $89.7\%(\pm 0.5)$ with total privacy budget $\epsilon = 5.8$ using our new privacy analysis, whereas, [24] (together with the standard strong composition theorem) achieves the same accuracy with a total privacy budget of $\epsilon = 18.3$.

**Computation resources:** For our experiments, we used a server which has 6 Nvidia RTX2080Ti GPU's and Intel Xeon Gold 6230 CPU @ 2.10GHz CPU's. The longest epoch time is $400$ seconds for training on MNIST.