# OpenReview forum: "Renyi Differential Privacy of The Subsampled Shuffle Model In Distributed Learning"
_NeurIPS.cc/2021/Conference — NeurIPS 2021 Poster_

### Official Review · Reviewer_HpvA · 2021-07-06

**Rating:** 6
**Confidence:** 3

**Summary:**

The focus of the paper is on Rényi differential privacy (RDP) in the shuffle model of DP. The main result in the paper is to introduce (at least somewhat loose) upper and lower RDP bounds for an arbitrary subsampled discrete mechanisms in the shuffle model. I think this is a novel, if fairly limited contribution.

**Limitations And Societal Impact:**

The limiting assumptions are stated clearly-enough. I do not think this paper needs to specifically discuss societal impact.

**Main Review:**

-- after rebuttals & discussion:
I still think this is a borderline paper and I therefore keep my score unchanged. I agree with the authors that the proofs have some novelty, but the paper also has some clear weaknesses (mainly the close relationship with the existing arXiv paper, looseness of the bounds, and the gap between the upper and the lower bound).
--

The authors consider privacy accounting of arbitrary discrete mechanisms with subsampling in the shuffle model of DP, a recently introduced distributed setting assuming a trusted shuffler besides the dataholders and the analyst or central server. As the main results of the paper, the authors introduce an upper and a lower bound for RDP. The bounds are not tight, and there is a significant gap between the bounds. However, the upper bound does improve on the existing methods.

The main reservations I have about this paper is that it seems almost as a small slice cut off from the more extensive paper on archive (and cited in the current paper, Girgis et al. 2021: On the renyi differential privacy of the shuffle model), and the contribution in this paper alone seems limited. Without properly checking the proofs, it also seems like they are mostly simple modifications on the existing proofs of the arXiv paper, combined with some techniques from an earlier subsampled RDP paper (Wang et al. Subsampled Rényi Differential Privacy and Analytical Moments Accountant).

All in all, the paper does make some clear improvement on the current methods, but it seems to be fairly limited in scope.

Questions/comments for the authors:
1) Since most of the proofs follow closely (sometimes almost verbatim) the proofs of the arXiv paper, I think this needs to be stated clearly also in the main paper for each Thm (currently this is mentioned in the Supplement).

Minor comments
* Add bibliography to the Supplement as the citation numbers differ from the main text.
* You could add a paragraph on alternative approaches to privacy accounting (maybe in the extended related work?) to dispel the idea that RDP is the only possible approach (compare e.g. privacy loss distributions or Gaussian DP). However, since these haven't been used in the shuffle model (as far as I know), I do not think this needs to be too extensive.
* Fix the spelling of Rényi, pretty please.

**Time Spent Reviewing:**

4

---

> ### Author Response · Authors · 2021-08-10
> **Difference from other mentioned works:**
>
> There are many differences in the proofs between [GDD+21: On the Renyi Differential Privacy of the Shuffle Model, by Girgis et al., 2021, on arXiv; BKW19: Subsampled Renyi Differential Privacy and Analytical Moments Accountant, AISTATS19] and the current paper, as explained below.
>
> Before going into the details, we would like to emphasize that one of the main objectives of our paper is getting a tighter trade-off between privacy and convergence of learning algorithms (via SGD) for smooth convex objectives in the shuffle privacy framework, which none of the above-mentioned papers do, neither theoretically not empirically. Furthermore, when we apply the techniques from the above papers for SGD in numerics, we observe significant gains in privacy-learning performance by our method. Our numerics show that our theoretical results achieve a significant gain in comparison to the existing results. Furthermore, we numerically evaluate the proposed privacy-learning performance for training the MNIST dataset for $\ell_{\infty}$ norm clipping and $\ell_2$ norm clipping. We show that we can achieve a reasonable learning performance with small central eps-DP. As far as we know, the best-known result on training MNIST achieves an accuracy of $87.9\%$ with $\epsilon=10$ for $\ell_{\infty}$-norm clipping (Girgis et. al. AISTATS21). In this paper, we achieve an accuracy of $90\%$ for $\epsilon= 4.8$ with $\ell_{\infty}$ norm clipping. This gain, we believe, is non-trivial, as constants matters in differential privacy.
>
> - Difference in the upper bound proof: Our proof consists of three steps as follows. First, we reduce the RDP of the subsampled shuffle model to the ternary DP analysis of non sub-sampled shuffle model (Lemma 2) — this step was not there in [GDD+21] but was observed in [BKW19] for non-shuffled mechanisms. Second, we reduce the ternary DP of the shuffle model for general triple neighboring datasets to the neighboring datasets having special forms (Theorem 5). The novel parts in this step are as follows: (i) proving some properties on the ternary DP such as its joint convexity (see Lemma 3) and monotonicity property of the ternary divergence (Lemma 5). These lemmas are new and do not exist in [GDD+21] and [BKW19]. Furthermore, these lemmas are crucial to establish the reduction. Third, we analyze the ternary DP for the triple of neighboring datasets with a special structure (Theorem 6). The new technique in this step is to reduce the ternary divergence of special triple neighboring datasets to RDP divergence of a pair of neighboring datasets (See corollary 1).
>
> Observe that the last two steps in our proof are for the ternary DP analysis that cannot be obtained directly from [GDD+21] and/or [BKW19]. In contrast, [GDD+21] bounds the RDP of the non sub-sampled shuffle model ($\gamma=1$) in two steps by reducing the RDP of a general pair of neighboring datasets to the RDP of a special form of neighboring datasets, and then the authors bound the RDP of this special form of neighboring datasets.

---

> ### Author Response · Authors · 2021-08-10
> **Fixing bibliography to the Supplement:**
>
> We will fix this.

---

> ### Author Response · Authors · 2021-08-10
> **Alternative approaches to privacy accounting:**
>
> We are happy to add a small discussion about the different approaches for composition in the literature review.

---

> ### Author Response · Authors · 2021-08-10
> **Spelling of Renyi (put the accent on e):**
>
> We will fix this.

---

> ### Author Response · Authors · 2021-08-10
> **arXiv paper mention in the main text:**
>
> We will incorporate this.

---

### Official Review · Reviewer_65kL · 2021-07-15

**Rating:** 7
**Confidence:** 3

**Summary:**

This paper follows the line of research aiming at tightening the privacy analysis of existing differentially private learning schemes. The authors consider the distributed learning framework, where a group of clients collaborate within a parameter-server architecture to learn a joint ML model under privacy (against the server). In this context, the paper analyzes the differential privacy guarantees obtained by  combining (i) subsampling of the clients (ii) local randomization at the client level and (iii) shuffling clients’ responses before sending them to the server. The authors provide new (upper and lower) bounds on the privacy guarantee of the above scheme for discrete local randomizers.  To do so, they rely on some well-known result from differential privacy (especially Renyi differential privacy), but also devise some interesting new proof schemes. The numerical comparison with existing works show that in some parameter regimes, the proposed bound considerably improves existing bounds studying the same framework.

**Limitations And Societal Impact:**

I believe the authors adequately addressed both the limitations and social impact of their work.

**Main Review:**

Relevance: The problem of tightening the privacy analysis of the distributed learning paradigm is an active and important line of research that is perfectly suited to a major ML conference like NeurIPS.

Clarity: The paper is overall clear and well written. I like that the authors provided sketch of proof and discussions on their results while presenting them (i.e., in the main body of the paper).

Quality: I did not have the time to go over all details of the proofs (hence my low confidence), but from my level of reading, both the technical and empirical contents look sound.  I just list below a couple of questions to the authors.
- The problem formulation (line 139) considers only one data point per user, which is quite restrictive. The authors briefly explained in the limitations paragraph (line 90) that it might not be trivial to generalize their results to arbitrary numbers of data points per user, and claim that their results are close ‘in spirit’ to uniform sampling. However it is unclear to me if the present results are simple to generalize even in the uniform sampling case. Could the authors clarify this point for me?
- Numerical results presented in Figure 2 and 3 presents some considerable improvements in the privacy/accuracy trade-off compared to previous works in certain parameter regimes. However in some other cases (e.g., in the extreme privacy/ low accuracy regime on the left of Figure 3), the comparison seems at the disadvantage of the new technique. Do the authors have some (theoretical or empirical) insight on what parameter regimes are the best suited to their analysis?

Originality: I think that the paper is not very original, but its theoretical contribution seems sufficiently significant to me. I think the authors provided an analysis of the subsampled + shuffle framework by making a good use of previously known results and by devising new proof schemes when needed. In particular, I believe that the reduction in Section 6 could be of independent interest in future works.

**Time Spent Reviewing:**

5 to 7

---

> ### Author Response · Authors · 2021-08-10
> **Arbitrary number of data points per client:**
>
> When each client has more than one data point, then, as we explain below, the extension of our sub-sampling method is non-trivial to analyze; however, there is an alternative sub-sampling strategy, described subsequently, that could be analyzed with more technical work.
>
> The straightforward sub-sampling analysis is non-trivial: Suppose there are $n$ clients, each having $m>1$ data points, and the algorithm first picks $k$ clients uniformly at random (for client sampling) and then each selected client, in turn, picks $s$ data points uniformly at random (for mini-batch SGD), then it is not hard to see that the resulting sub-sampling of $ks$ data points from the total $nm$ data points is not uniform. Therefore, the existing privacy by sub-sampling results (which are inherently for uniform sub-sampling) is not directly applicable here. This requires a new proof technique for reducing the RDP analysis of non-uniform sub-sampled shuffle mechanisms to the ternary-DP analysis of (non sub-sampled) shuffle mechanisms (i.e., extending Lemma 2 to the non-uniform sub-sampling case), which is non-trivial. Everything else will remain the same.
>
> An alternate sub-sampling strategy, which resembles the uniform sub-sampling and is amenable to analysis is the following: Each of the $n$ clients uniformly at random samples $s$ data points and sends their gradients;  the shuffler receives $ns$ gradients, out of which it selects $ks$ gradients uniformly at random and shuffles them. This can also be analyzed with a little more technical work.

---

> ### Author Response · Authors · 2021-08-10
> **What parameter regimes are the best suited to the new analysis?**
>
> Our new technique achieves gain when $\epsilon_0$ is small and/or the number of rounds $T$ is large. The reason is that there is a gap of $\mathcal{O}(e^{\epsilon_0})$ between the lower and the upper bounds on the RDP of the sub-sampled shuffle mechanism, and hence, our upper bound is useful for an arbitrary number of rounds $T$ when $\epsilon_0$ is small. Furthermore, the analysis of the RDP is useful for composition when the number of rounds $T$ is large even when $\epsilon_0$ is large.

---

### Official Review · Reviewer_cNC6 · 2021-07-17

**Rating:** 6
**Confidence:** 3

**Summary:**

The paper studies discrete randomization mechanism in the subsampled Renyi DP model. They give a bound on the RDP of the subsampled shuffle mechanism and also give a lower bound.

**Limitations And Societal Impact:**

Yes

**Main Review:**

The biggest positive of the paper is the upper bound on subsampled renyi DP. The bound is concrete and can be implemented in practice to account for the privacy loss in the real world when we use shuffle DP. I verified the proof and it is correct to the best of my understanding.

I will next point out the cons against the paper.
1. The paper studies pure LDP mechanism whose range is a discrete set while one of the applications of the paper is SGD. The loss function in the SGD has the range over the real. This is a discrepancy which I do not understand. For example, for p=2, the mechanism of DJW samples a uniform point in the \ell_2 ball in the direction of the gradient with probability e^eps/(1+e^eps). This would always yield a point whose coordinates are real.
2. The proof technique in the paper is almost identical to that by Girgis et al. "On the Renyi Differential Privacy of the Shuffle Model." The only difference is that the reduction to the special case where they have to consider three databases of special form. This has to do with the proof technique introduced in Balle et al. analytic moment paper for subsampled RDP (BKW19). Even the observation that the randomization in the shuffle model can be seen as a weighted convolution of two probability distributions was made in FMT20. I fail to see what it new in this paper. I would appreciate if the authors can point exactly what is new in the paper or whether it is just the adaption of Girgis et al. to BKW19 and FMT20.
3. Since the expectation is not always on a single distribution, I would suggest the authors to make it explicit. For example, when they prove the joint convexity of ternary divergence, what is the expectation over in each of the term? Is it over R_a, R_0 and R_1, respectively? Then I believe you need to use Jensen, the joint convexity of bivariate function, and Fubini.

Finally comparing the result with the Girgis et al.'s main result, setting gamma=1 in Theorem 1 does not yield Theorem 1 in Girgis et al. As a result, there is a big discontinuity near 1 when we map RDP guarantee with subsampling rate.

Unless these concerns are not addressed explicitly, I am afraid the paper is a clear reject for me. I am currently keeping the score high enough in an optimistic hope that the authors address it clearly and concisely giving me reasons to increase my score.

**Time Spent Reviewing:**

10

---

> ### Author Response · Authors · 2021-08-09
> **Discrete and continuous mechanisms:**
>
> Although the loss function and the gradients are real-valued for learning, the gradient compression/quantization techniques in FL (e.g., QSGD, VqSGD, signSGD, etc.) produce gradients whose components can only take finite values (i.e., they are not real-valued), and we can still use them in the SGD procedure for minimizing real-valued loss functions. This is because the descent step in SGD does not require the stochastic gradients to be real-valued; it only requires them to be unbiased and have bounded variance (or bounded second moment). Therefore, for quantization, it suffices to show that the quantized stochastic gradients are unbiased and have bounded variance (w.r.t. the true real-valued gradients). This is exactly what we prove in the paper that the quantized gradients (which are the output of the LDP mechanisms) are unbiased and have bounded variance; please see the proof of Theorem 3 in Appendix F. So, we believe that there is no discrepancy here. For example, for $p=2$, we can quantize the LDP mechanism of DJW to discretize its output (without hurting the privacy, as quantization is post-processing which retains privacy) and can use it in SGD procedure. Hence, our results can be used for real outputs as well. We would like to note that there is a lot of recent work that studied discrete LDP mechanisms for learning and shuffling, e.g., Chen et al. (NeurIPS20), Girgis et al. (JSAIT20) Feldman et al. (Arxiv21), and Kairouz et al. (ICML21). Thus, we believe that our results would be helpful for a broad range of mechanisms.

---

> ### Author Response · Authors · 2021-08-10
> **Our Contributions:**
>
> [GDD+21] On the Renyi Differential Privacy of the Shuffle Model, by Girgis et al., 2021, on arXiv.
>
> [BKW19]: Subsampled Renyi Differential Privacy and Analytical Moments Accountant, AISTATS19.
>
> Before discussing the differences in the proofs between [GDD+21], [BKW19] and the current paper, we would like to emphasize that one of the main objectives of our paper is getting a tighter trade-off between privacy and convergence of learning algorithms (via SGD) for smooth convex objectives in the shuffle privacy framework, which none of the above-mentioned papers do, neither theoretically nor empirically. Furthermore, when we apply the techniques from the above papers for SGD for numerics, we observe significant gains in privacy-learning performance by our method. This, we believe, is a non-trivial and important application of our new theoretical results. We will discuss this after describing the major theoretical differences between the current paper and the above-mentioned ones.
>
> - Difference in the upper bound proof: The proof strategy in [GDD+21] consists of two steps, whereas, our proof has three steps. In [GDD+21], the first step is to reduce the RDP analysis for arbitrary neighboring datasets to the RDP analysis for neighboring datasets with special structures, and then the second step is for computing the RDP for the latter case. The three proof steps in the current paper along with differences from [GDD+21] and new technical parts are described below:
>
>
> $\bullet$ The first step is to reduce the RDP analysis for sub-sampled shuffle mechanisms to the ternary DP analysis of non sub-sampled shuffle mechanisms (Lemma 2) — this step was not there in [GDD+21], but was observed in [BKW19] for non-shuffle mechanisms, whereas we do it for shuffle mechanisms.
>
> $\bullet$ The second step is to reduce the ternary DP analysis for arbitrary triple of neighboring datasets to the neighboring datasets having special structures (Theorem 5). The new technical parts in this reduction are as follows: (i) Adapting the proofs of the required lemmas from Renyi divergence (in [GDD+21]) to the ternary divergence; and (ii) proving joint-convexity of the ternary divergence (see Lemma 3), which is new and was not shown in [GDD+21] and [BKW19]; and (iii) a monotonicity property of the ternary divergence (Lemma 5), which uses the joint convexity established in Lemma 3.
>
> $\bullet$  The third step is to analyze the ternary DP for the triple of neighboring datasets with a special structure (Theorem 6). The new technical part in this step is the reduction of computing the ternary divergence (of a triple of datasets) to computing the Renyi divergence (of pairs of datasets), and then use results from [GDD+21] to bound the Renyi divergence (see Corollary 1).
>
> Note that both the above steps in [GDD+21] are for the RDP analysis, whereas, the last two steps in the current paper are for the ternary DP analysis, which, as explained above, do not directly follow from the analysis of [GDD+21].
>
> - Experimental details: Our numerics show that our theoretical results achieve a significant gain in comparison to the existing results when evaluating the proposed privacy-learning performance for training on the MNIST dataset for $\ell_{\infty}$-norm clipping and $\ell_2$-norm clipping. We show that we can achieve a reasonable learning performance with small central $\epsilon$-DP in comparison to the state-of-the-art result (Shuffled Model of Differential Privacy in Federated Learning, by Girgis et al., AISTATS'21). In particular, Girgis et al.'21 achieve an accuracy of $87.9\%$ with $\epsilon=10$ for $\ell_{\infty}$-norm clipping, whereas, we achieve an accuracy of $90\%$ for $\epsilon= 4.8$ with the same $\ell_{\infty}$-norm clipping. This gain, we believe, is non-trivial, as constants matters in differential privacy.

---

> ### Author Response · Authors · 2021-08-10
> **Expectation in the joint-convexity of ternary divergence in Lemma 3:**
>
> Yes, the expectations are over $R_a,R_0$, an $R_1$, respectively -- we will write this explicitly; thanks for pointing this out. For the proof, we do not think we need Fubini or Jensen's as we do not exchange expectation and integrations/summations. The two inequalities in eqn (14) are obtained from the convexity of the absolute function and the bivariate function, respectively. Note that eqn (14) is point-wise; we then integrate (14) to get the desired inequality in (12).

---

> ### Author Response · Authors · 2021-08-10
> **Discontinuity at $\gamma=1$:**
>
> In general, the difference between the RDP in Theorem 1 of our work at $\gamma=1$ and the RDP in Theorem 1 in Girgis et. al.'21 is in the coefficients of each term in the summation, where there is a multiplicative gap of $2^j$ in the $j$th term. This difference comes because the proof techniques are different -- we bound the ternary divergence and then transform it to RDP, while Girgis et. al.'21, bound the RDP directly. Although we lose some constants when transforming the ternary DP to RDP, the final bound that we obtain is still very useful for sampling when $\gamma<1$, as we show numerically in Section 4.
>
>
> Even at $\gamma=1$, for some practical setting of parameters, we can show that our bound is comparable with the bound of Girgis et al.'21 (On the Renyi differential privacy of the shuffle model, arXiv'21). For example, when $\gamma=1$, $\epsilon_0=2$, number of clients $n=10^6$, and RDP order $\lambda=100$, our RDP analysis gives $\epsilon=0.0616$, while the RDP analysis of Girgis et. al.'21 gives $\epsilon=0.0234$. Thus, for this example, the multiplicative gap between our bound and that of Girgis et. al.'21 is around $2.5$.

---

> > ### Comment · Reviewer_cNC6 · 2021-09-01
> > **Discontinuity issue in practice**
> >
> > Thanks for doing the calculation. Unfortunately, I do not completely agree with it that it is fine to have a multiplicative gap of order greater than $1+\alpha$ for arbitrary small $\alpha$. The reason being in practice, consider two experiments. In one experiment, one decides to do a gradient descent taking the entire training data in every sample. I would do accounting based on Girgis et al. In the second, I do almost a gradient descent, but take one training data out of my training. In that case, my subsampling rate is $(n-1)/n$ and I would have to do accounting based on your result. I would get a higher privacy loss in the second experiment, which would be hard to explain to a practioner!

---

> > > ### Author Response · Authors · 2021-09-02
> > > **Discontinuity issue in practice**
> > >
> > > We thank the reviewer for the question after reading our first response. We would like to take this opportunity to further clarify our response given this question. In practical federated learning, which was the motivation of our work, the (client) sampling parameter $\gamma$ is small; for instance, the description of cross-device setups (see reference [0] at the end of the response) in large-scale deployments of federated learning, one could have 100s of thousands to millions of clients (devices) and only a few thousand would be sampled yielding a very small $\gamma$ (see Table 2 on Page 8 in [0] for reference). Moreover, in other established works, in training MNIST, CIFAR10, or Fashion-MNIST datasets, the sampling parameter $\gamma$ is less than 1/6 (e.g.,[1,2,3]). Our analysis of RDP of the subsampled shuffled mechanism achieves a significant gain in this important regime. Furthermore, we show some numerics in Section 4 showing that we achieve an accuracy of $90\%$ with a total privacy budget $\epsilon=2.91$ for training MNIST dataset, while the best privacy analysis in the literature achieves an accuracy of $90\%$ with a total privacy budget $\epsilon=4.82$.
> > >
> > >
> > > We want to emphasize another point, one can combine the RDP analysis of Girgis et al. [GDDSK21] along with the sub-sampling RDP results [BKW19] to obtain sub-sampled privacy analysis for all $\gamma\in(0,1]$ (a point we made in footnote 4 on page 3 of our submission). We compare our results to this baseline approach in the paper and also in the regime suggested by the reviewer below. However, we would like to reiterate that our paper gives a direct analysis for the RDP of the subsampled shuffle mechanism that outperforms [GDDK21]+[BWK19] in important regimes (especially small $\gamma$ as shown). Now, for the specific regime raised by the reviewer, where $\gamma=1-1/n$, consider, number of clients $n=10^6$, $\epsilon_0=2, \gamma=1-1/n$, and the RDP order $\lambda=100$, we get that our RDP analysis gives $\epsilon=0.0616$, while the RDP analysis of [GDDSK21]+[BKW19] gives $\epsilon = 0.712$. This shows that even in the regimes mentioned, we can get significant gains. Finally, it is clear that one can analyze our direct method and the composition of [GDDSK21]+[BWK19] (as described above and in footnote 4 of the submission), and use the better of the two. What we have demonstrated in the numerics is that the direct analysis can give significant benefits in the important parameter regimes. As a conceptual idea, we do not think this is difficult to explain to a practitioner, as we will know the regime where we need to operate and we can give the privacy guarantees obtained by the methods, choosing the appropriate one to use. The message is that in most important regimes the direct method advocated in this submission is much better.
> > >
> > >
> > > [0] Kairouz et al., “Advances and Open Problems in Federated Learning”, arXiv: 1912.04977, 2019.
> > >
> > > [1]  Erlingsson, Úlfar, et al. "Encode, shuffle, analyze privacy revisited: Formalizations and empirical evaluation." arXiv preprint arXiv:2001.03618 (2020).
> > >
> > > [2]  Abadi, Martin, et al. "Deep learning with differential privacy." Proceedings of the 2016 ACM SIGSAC conference on computer and communications security. 2016.
> > >
> > > [3]  Papernot, N., Chien, S., Song, S., Thakurta, A., & Erlingsson, U. (2019). Making the shoe fit: Architectures, initializations, and tuning for learning with privacy.

---

> > > > ### Comment · Reviewer_cNC6 · 2021-09-02
> > > > **thanks**
> > > >
> > > > Thanks for the quick response. I agree with the authors that their result gives better results for various regimes of epsilon and RDP parameters that naively combining Girgis et al. and Balle et al. The authors have discussed this in the submission, too.
> > > >
> > > > My question was more in regards to the continuity of results when we move from $\gamma = 1-1/(poly(n))$ to $\gamma = 1$. I believe Girgis et al. (shuffle model privacy) gives a tighter bound compared to this paper when the subsampling rate is set to 1. I do like the problem studied in the paper and I am just trying to understand why is there slack in this regime. Do the authors think it is the artifact of the analysis? Maybe, in FL where 1 million users participate, you would mostly be concerned with small $\gamma$, but there might be other applications of FL where you need to do entire gradient descent. This might be a more hypothetical situation, but say one is trying to get some health-related model with a small number of clients and we are continually monitoring and updating the model as we continually observe them every day. One day, one client failed to put his phone on charge, so we do not get their data. The sampling rate moves from $\gamma=1$ to $\gamma = 1 - 1/n$ and the privacy accounting of our system would not remain robust to such a change.

---

> > > > > ### Author Response · Authors · 2021-09-02
> > > > > **Thank you and further clarifications**
> > > > >
> > > > > We thank the reviewer for the positive comments about our work and the problem studied.
> > > > >
> > > > > We want to emphasize that when $\gamma$ is even slightly less than one, say, $\gamma=0.99999$, our bound beats the current best bound (which is [GDDSK21]+[WBK19]) by a significant margin as demonstrated in the last response).
> > > > > As stated in our earlier response as well, for the specific value of $\gamma=1$ our bound is different from the bound of [GDDSK21], and we believe it is not fundamental but an artifact of the different analysis techniques. As we had explained the reason for slack at $\gamma=1$ in the first response, the difference between the RDP in Theorem 1 of our work at $\gamma=1$ and the RDP in Theorem 1 in [GDDSK21] comes because the proof techniques are different -- we bound the ternary divergence and then transform it to RDP, while [GDDSK21] bound the RDP directly. As argued earlier as well, the final bound that we obtain is still very useful in important regimes when $\gamma<1$, as we also show numerically in Section 4. We also want to re-emphasize that our bound significantly improves upon the existing bounds when there is an actual sub-sampling (i.e., $\gamma<1$) which also allows us to get a significantly better privacy-learning tradeoff than the existing methods (as shown in Figure 3) in the important parameter regimes, which was the main motivation of our work.
> > > > >
> > > > > We would also like to point out that in your example scenario, the sub-sampling rate $\gamma$ is changing over iterations, and none of the existing results (including ours) that we know of can analyze the RDP of sub-sampled mechanisms with arbitrarily varying sub-sampling rates. One way to handle this is to assume an upper bound on the sub-sampling rate across all iterations, and use that while evaluating the bound. When we know the $\gamma$ and other parameters, we can have explicit provable privacy bound and analysis. Analyzing privacy for arbitrarily varying sub-sampling rates is a challenging unstudied problem (to the best of our knowledge).
> > > > >
> > > > > We hope that we have addressed all the concerns raised earlier. We would greatly appreciate it if you could reevaluate your rating based on our responses. Please let us know if you have any more concerns and we would be happy to answer.

---

> > > > > > ### Comment · Reviewer_cNC6 · 2021-09-02
> > > > > > **thanks**
> > > > > >
> > > > > > Thanks to the authors for clarifying in more detail. The engagement was very helpful in me understanding the paper a bit better.

---

> > > > > > > ### Author Response · Authors · 2021-09-02
> > > > > > > **Thank you for increasing the score**
> > > > > > >
> > > > > > > Thank you for your positive response and also for increasing your rating; we appreciate it. We are glad to have answered the questions raised; hopefully, they clarified them.

---

### Official Review · Reviewer_jtYS · 2021-08-02

**Rating:** 7
**Confidence:** 3

**Summary:**

This paper studies the problem of bounding the privacy loss of the sub-sampled shuffle model, motivated by the real-world application of federated learning with differential privacy.

The broad type of mechanism studied in this paper is as follows:\
Setup:
- Consider dataset $(d_1, \ldots, d_n)$ where client #$i$ has the datapoint $d_i \in \mathcal{X}$.
- Consider any $\epsilon_0$-locally differentially private mechanism $\mathcal{R} : \mathcal{X} \to [B]$.

(One round of) the mechanism is as follows:
- $k$ out of $n$ clients are chosen. Each chosen client $i$ send $\mathcal{R}(d_i)$ to the _shuffler_.
- The shuffler randomly permutes the received responses from the $k$ chosen clients and sends it to a central server, which can post-process the same in an arbitrary manner.

The goal is to know, for any given $\delta$, what is the smallest $\epsilon$ for which the above mechanism is $(\epsilon, \delta)$-differentially private (DP) : this $\epsilon$ can depend on $\epsilon_0$, $k$, $n$ and $\delta$.

One approach for analyzing the privacy loss is to compose (in some way) the privacy loss guarantees of each of the two steps : sub-sampling and shuffling. One such way is to compose the Renyi differential privacy guarantees of the two steps (since Renyi-DP usually gives better bounds than the standard "advanced composition"). This paper instead takes a direct approach to bounding the Renyi-DP loss, and also demonstrates empirically that doing so, does better than performing any kind of composition of the two steps.

The paper uses the recently introduced technique of _ternary-$|\chi|^{\alpha}$-differential-privacy_ to prove the theorems.

**Ethical Concerns:**

I do not have any ethical concerns regarding this paper.


**Limitations And Societal Impact:**

The paper raises good points about the potential negative societal impacts of this work: namely that private learning is inherently at odds with checkability of fairness criteria (since the data is private, it can be hard to directly check whether the data was biased or not). It also points out that a small subset of adversarial users could contribute malicious data to the learning algorithm, thereby leading to unfavorably learnt models.

These issues are not specific to the results in this paper, but are broadly applicable to learning with differential privacy.


**Main Review:**

The problem studied in this paper is practically very relevant and timely, as federated algorithms are being deployed in practice at a massive scale and privacy is a mounting concern. The shuffle model is emerging as an important middle ground between the central and local models of differential privacy.\
It is important to have as tight a guarantee on the privacy parameters as possible, so as to have good utility while preserving privacy. Even constant factor improvements in estimating the privacy loss can be practically relevant.

The main contribution in the paper is the direct analysis of the privacy loss in the subsampled shuffle model. The main technique used is the ternary $|\chi|^{\alpha}$-DP that was introduced in a recent work of [Wang et al. 2019].

This direct analysis is empirically shown to yield better bounds than composing the privacy guarantees that have been obtained previously for sub-sampling and for shuffling separately (different kinds of compositions are considered, and the bound proved in this paper is significantly better than these). Moreover, the paper also proves a lower bound on the Renyi-DP loss of the sub-sampled shuffled mechanism : this is also useful to know in practice.

#### Significance and originality:

While the direct analysis in the paper is technically challenging and involved, I feel it is somewhat incremental over the prior work of [Wang et al. 2019].\
However, the improved bounds could be highly impactful in practice.

#### Quality of the paper:

The paper is well-written overall, and contributions are clearly stated.\
While I did not carefully verify all the proof details, they make sense at a high level.

#### Minor Typo:

- Page 3 (Defn 1). Since this definition is for a fixed $\lambda$, I think $(\lambda, \epsilon(\lambda))$ should be simply $(\lambda, \epsilon)$.

**Time Spent Reviewing:**

5

---

> ### Author Response · Authors · 2021-08-08
> **$\lambda$ in Def 1:**
>
> Thank you for pointing it out; we will fix this in Eq. (1) in Def. 1.

---

### Decision · Program_Chairs · 2021-09-27

**Decision:**

Accept (Poster)

**Comment:**

There is a consensus that the paper provides an interesting theoretical contribution to an important and timely problem, the shuffle model being an actively studied model of privacy that is also increasingly studied in federated learning (FL).

The authors' result on analyzing the privacy loss in the subsampled shuffle model and the accompanying empirical evaluation are valuable contributions towards tightening the privacy/utility trade-offs of DP-SGD in a natural FL setting.

However, there is also a consensus that the paper has a somewhat limited technical novelty, with proofs very similar to ones in recent work. Also, some concerns are raised regarding the remaining gap between the obtained lower and upper bounds.

That said, the reviewers are in agreement in leaning towards acceptance.